# Metabolic stress is a primary pathogenic event in transgenic *Caenorhabditis elegans* expressing pan-neuronal human amyloid beta

Emelyne Teo[1,2], Sudharshan Ravi[3,4], Diogo Barardo[2,5], Hyung-Seok Kim[2], Sheng Fong[6], Amaury Cazenave-Gassiot[5,7], Tsze Yin Tan[8], Jianhong Ching[8], Jean-Paul Kovalik[8], Markus R Wenk[5,7], Rudiyanto Gunawan[3], Philip K Moore[9], Barry Halliwell[5], Nicholas Tolwinski[2], Jan Gruber[2,5]*

[1]NUS Graduate School for Integrative Sciences and Engineering, National University of Singapore, Singapore, Singapore; [2]Science Division, Yale-NUS College, Singapore, Singapore; [3]Department of Chemical and Biological Engineering, University of Buffalo, Buffalo, United States; [4]Department of Chemistry and Applied Biosciences, Institute for Chemical and Bioengineering, ETH Zurich, Zurich, Switzerland; [5]Department of Biochemistry, National University of Singapore, Singapore, Singapore; [6]Geriatric Medicine Senior Residency Programme, SingHealth Duke-NUS Academic Medical Centre, Singapore, Singapore; [7]Singapore Lipidomics Incubator, National University of Singapore, Singapore, Singapore; [8]Cardiovascular and Metabolic Disorders Programme, Duke-NUS Medical School, Singapore, Singapore; [9]Department of Pharmacology, National University of Singapore, Singapore, Singapore

*For correspondence:
yncjg@nus.edu.sg

Competing interests: The authors declare that no competing interests exist.

**Abstract** Alzheimer's disease (AD) is the most common neurodegenerative disease affecting the elderly worldwide. Mitochondrial dysfunction has been proposed as a key event in the etiology of AD. We have previously modeled amyloid-beta (Aβ)-induced mitochondrial dysfunction in a transgenic *Caenorhabditis elegans* strain by expressing human Aβ peptide specifically in neurons (GRU102). Here, we focus on the deeper metabolic changes associated with this Aβ-induced mitochondrial dysfunction. Integrating metabolomics, transcriptomics and computational modeling, we identify alterations in Tricarboxylic Acid (TCA) cycle metabolism following even low-level Aβ expression. In particular, GRU102 showed reduced activity of a rate-limiting TCA cycle enzyme, alpha-ketoglutarate dehydrogenase. These defects were associated with elevation of protein carbonyl content specifically in mitochondria. Importantly, metabolic failure occurred before any significant increase in global protein aggregate was detectable. Treatment with an anti-diabetes drug, Metformin, reversed Aβ-induced metabolic defects, reduced protein aggregation and normalized lifespan of GRU102. Our results point to metabolic dysfunction as an early and causative event in Aβ-induced pathology and a promising target for intervention.
DOI: https://doi.org/10.7554/eLife.50069.001

## Introduction

Alzheimer's disease (AD) is a debilitating neurodegenerative disease, that is clinically characterized by the formation of amyloid-beta (Aβ) plaques and aggregates of hyperphosphorylated tau protein in the brain (*Mirra et al., 1991*). Even though AD is primarily a neuronal disorder, perturbations in

mitochondrial functions including energy metabolism have consistently been observed not only in the brain (*Gibson et al., 1998*; *Gibson et al., 1999*; *Müller et al., 2010*) but also in non-neuronal cells derived from AD subjects, including in fibroblasts and platelets (*Bosetti et al., 2002*; *Blass and Gibson, 1992*; *Cardoso et al., 2004*; *Curti et al., 1997*; *Parker et al., 1994*; *Swerdlow, 2012*). These findings form part of an emerging story that there is an important metabolic component to the etiology of AD (*de la Monte and Wands, 2008*), and that these metabolic defects may precede Aβ aggregate formation (*Yao et al., 2009*; *Ye et al., 2012*). The metabolism-related hypothesis of AD therefore posits that AD is, in part, mediated by impairments to the brain's insulin response, which promotes oxidative stress and inflammation, similar to that of diabetes (*de la Monte and Wands, 2008*; *McClean et al., 2011*). Moreover, intranasal insulin treatment has been shown to ameliorate AD pathology in a transgenic rat model and to improve mild cognitive impairment (MCI) in patients (*Craft et al., 2012*; *Guo et al., 2017*; *Stanley et al., 2016*; *Craft et al., 2017*).

Several animal studies have confirmed that oxidative stress, mitochondrial dysfunction and metabolic alterations are early events in the pathophysiological progression of AD. Energy deficits, reduction in mitochondrial membrane potential, abnormal mitochondrial gene expression and increased oxidative stress have been observed early in transgenic AD mice (2–3 months of age) well before the appearance of Aβ plaques (*Reddy et al., 2004*; *Hauptmann et al., 2009*). The Aβ-induced oxidative stress hypothesis further posits that Aβ, predominantly its oligomeric form, can directly induce oxidative stress by inserting itself into the lipid bilayer, subsequently leading to lipid peroxidation and protein oxidation (*Swomley and Butterfield, 2015*; *Swomley et al., 2014*). Oxidative modification to specific enzymes involved in glucose metabolism has also been confirmed in brain of AD and MCI patients (*Butterfield et al., 2006*; *Reed et al., 2009*), further supporting the hypothesis that Aβ-induced oxidative stress contributes to defects in metabolism, impairs glucose metabolism, before leading to synaptic dysfunction and ultimately neuronal death in AD (*Butterfield and Halliwell, 2019*).

*Caenorhabditis elegans* (*C. elegans*) is an invertebrate organism with unique advantages as a model organism. *C. elegans* has a short lifespan, a well-characterized nervous system, and is therefore routinely used for studies in developmental biology, neurobiology, aging and mitochondrial biology (*Gruber et al., 2015*; *Ghosh-Roy and Chisholm, 2010*; *Corsi, 2015*; *Dancy, 2015*). The creation of transgenic AD strains expressing human Aβ-peptides in different cellular locations (muscle, neurons, or in subsets of neurons only) and under different promoters (temperature inducible or constitutive) has confirmed AD-like pathology with varying disease severity (*McColl et al., 2012*; *Wu et al., 2006*; *Link, 1995*; *Fong et al., 2016*; *Treusch et al., 2011*). In particular, temperature-inducible strains which, upon induction, express high levels of Aβ in muscle, exhibit obvious Aβ aggregates, rapid paralysis and die within 2–3 days (*Link, 1995*). Strains with a comparatively low-level of neuronal Aβ expression do not produce obvious aggregates and typically display milder effects, such as a reduction in egg-laying, progressive chemotaxis defects, neuronal loss and a 10–20% lifespan reduction (*Wu et al., 2006*; *Fong et al., 2016*; *Treusch et al., 2011*). Different strains therefore serve as useful tools to study different aspects of AD (late-stage, aggregate-driven pathologies in the more severely affected strains *vs.* early-stage AD in strains with later and milder phenotypes). In particular, it is important to study the early events occurring in AD pathogenesis, as interventions given at earlier stages of AD may be more likely to succeed in modifying disease progression.

To study the temporal sequence of events underlying Aβ-mediated toxicity, especially the early events affecting mitochondrial metabolism, we have created a novel transgenic *C. elegans* strain with constitutive pan-neuronal expression of low-levels of human Aβ$_{1-42}$ (GRU102). Similar to previous transgenic strains expressing Aβ in neurons, GRU102 display comparatively mild phenotypes including failure in chemotaxis, with neuromuscular and behavioral defects that become more pronounced with age. Remarkably, even though Aβ aggregates were detectable only in old GRU102 (day 12), we found defects in energy metabolism and significantly reduced activity in mitochondrial electron transport chain (ETC) complexes in GRU102 already when young (*Fong et al., 2016*). Our findings suggest that metabolic alterations are early events in AD pathogenesis, and this prompted us to further investigate why and how Aβ peptides cause mitochondria to fail, and what interventions might target early defects in the pathophysiological chain leading to AD.

Based on the metabolism hypothesis of AD, several studies have explored a new class of interventions, aiming at improving metabolic health, including the use of antioxidant and insulin-regulating

drugs in AD. For instance, we have previously shown that antioxidant treatment, using the mitochondrial-targeted antioxidant MitoQ, ameliorated depletion of the mitochondrial lipid cardiolipin and improved healthspan of another strain of transgenic AD nematodes with muscular expression of Aβ (CL2006) (*Ng et al., 2014*). Treatment using an antidiabetic drug, Metformin, has also recently been shown to delay Aβ-related paralysis, and improve neuronal transmission, in two transgenic AD strains, one with inducible muscular (CL4176) and one with neuronal Aβ expression (CL2355) (*Ahmad and Ebert, 2017*). However, the mechanisms underlying the beneficial AD effects of such metabolism-related drugs are still not fully understood.

Here, we examine how Aβ causes mitochondrial failure in AD, the mechanisms underlying the beneficial effects of Metformin, the metabolic consequences underlying Aβ expression and how they relate to protein aggregation in the GRU102 strain. We link Aβ expression to Tricarboxylic acid (TCA) cycle impairment and oxidative protein damage specifically in the mitochondria starting early in life (day 4). This precedes general protein aggregation as the latter only occurs late in life (day 12), suggesting the disruption in proteostasis is a secondary event likely induced by the early metabolic dysfunction. Metformin treatment normalized lifespan to that of untreated non-transgenic controls, increased stress resistance, reversed metabolic defects and prevented the increase in levels of protein aggregation in GRU102, suggesting that improvement in metabolism can also enhance proteostasis. Our results, taken together with other recent findings, suggest that disruption in mitochondrial function and redox homeostasis can induce protein aggregation, and targeting these processes can be beneficial in AD (*Ruan et al., 2017*; *Patel et al., 2017*; *Fang et al., 2019*; *Guerrero-Gómez et al., 2019*).

## Results

### Alterations in amino acid and tricarboxylic acid cycle metabolism in GRU102

GRU102 suffer from early and significant defects in energy metabolism (*Fong et al., 2016*). To determine which energy substrates were specifically affected in GRU102, we followed a targeted metabolomics approach, comparing intermediary metabolites involved in energy production between age-matched GRU102 and its transgenic controls (GRU101). We specifically determined levels of different Acylcarnitines (AC), Triacylglycerides (TAG), amino acids (AA) as well as of organic acids (OA), including TCA cycle intermediates.

AC are intermediates that are typically generated during the transport of activated long-chain fatty acids (FA) from the cytosol to mitochondria for FA oxidation (*Schooneman et al., 2013*). Even though most AC species are derived from FA, some AC species are formed through non-FA intermediates. For example, Propionylcarnitine (C3) and Isovalerylcarnitine (C5) are derived from the degradation of branched-chain AA while Acetylcarnitine (C2) is a common energy precursor derived from glucose and FA metabolism (*Schooneman et al., 2013*; *Rinaldo et al., 2008*). We found that levels of the common energy precursor C2 were significantly reduced in old GRU102 compared to age-matched GRU101 controls (*Figure 1A*, *Figure 1—figure supplement 1*), reflecting a low energy status in old GRU102. AA-derived AC species (C3-carnitine) was also significantly reduced in old GRU102 (*Figure 1A*), suggesting that AA metabolism was affected in old GRU102. By contrast, FA-derived AC species remained unchanged (*Figure 1A*), suggesting that FA metabolism remained intact even in old GRU102. These findings agree with the observation that levels of storage lipids, Triacylglycerides (TAGs), between GRU102 and GRU101 remained similar (*Figure 1B*) while lower total AA levels were observed in old GRU102 (*Figure 1C*).

AA can be metabolized to produce energy *via* glucogenic pathways where they are converted into pyruvate, alpha-ketoglutarate (aKG), succinyl CoA, fumarate or oxaloacetate, which can then participate in the TCA cycle (*Hayamizu, 2017*). To determine how changes in the AA profile of GRU102 relate to metabolic alterations, we classified individual AA by the glucogenic substrate that can be formed from them (oxaloacetate, pyruvate, aKG and succinyl CoA). We calculated the percentage of AA that forms each of the four glucogenic substrate by taking the sum of individual AA that forms the particular substrate divided by the total amount of AA. For Oxaloacetate, we took the levels of Aspartate over total AA levels; for Pyruvate, we took the sum of Alanine, Serine and Glycine over total AA levels; for aKG, we took the sum of Arginine, Histidine, Glutamine, Proline and

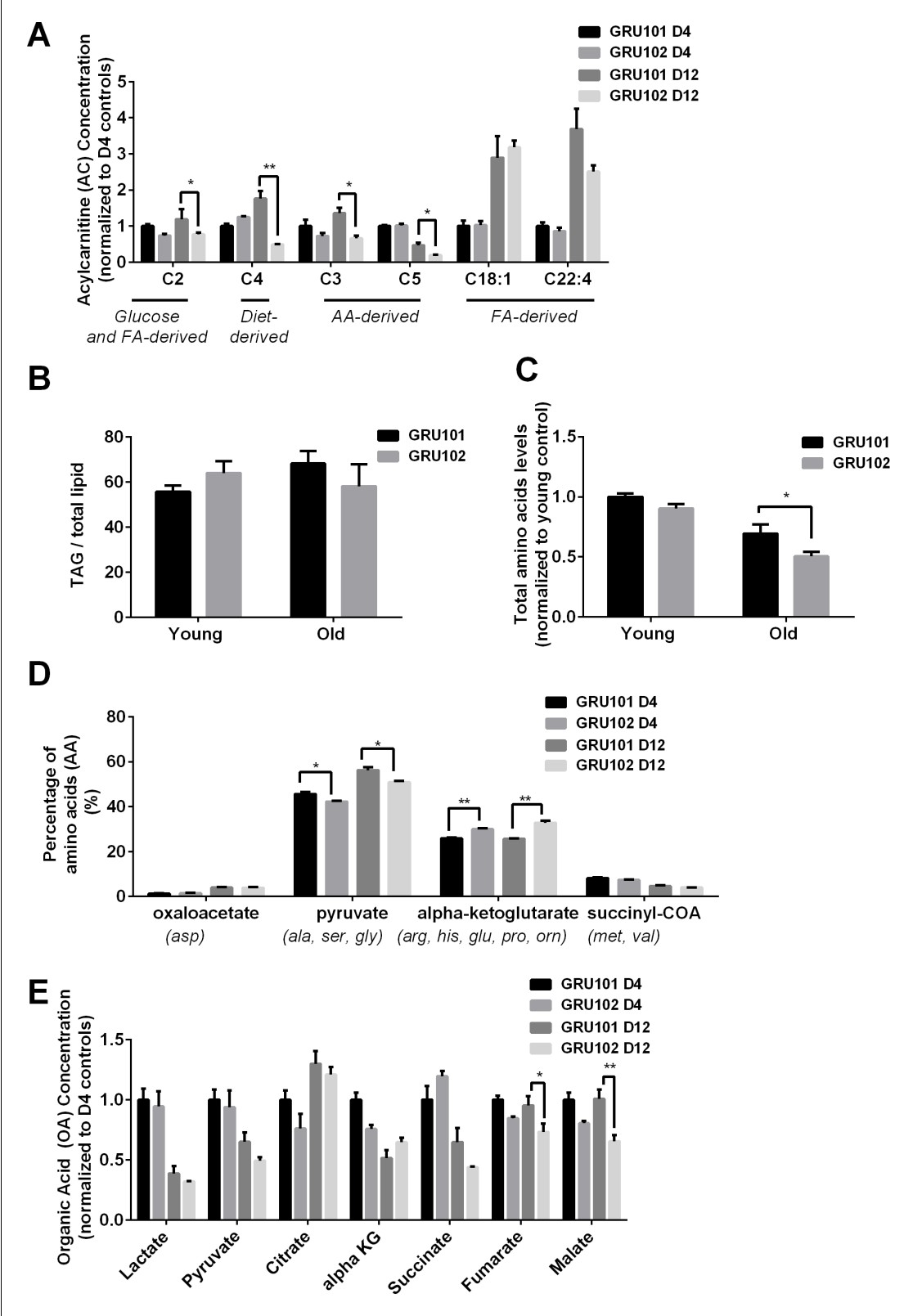

**Figure 1.** Metabolomics profile of GRU102 and its transgenic controls (GRU101). (**A**) Acylcarnitines (AC) profile, (**B**) Triacylglyceride (TAG), (**C**) Total Amino acids (AA) level, (**D**) Percentage glucogenic AA, computed as [sum of AA forming the particular glucogenic substrate/total AA levels]. (**E**) Organic acids (OA) concentration of GRU101 and GRU102. All values were normalized to respective protein concentration and then to young GRU101 (Two-way ANOVA Dunnett's multiple comparisons test, p<0.05, *; p<0.005, **; p<0.001, ***; n = 3 repeats per condition, with approximately 3000

*Figure 1 continued on next page*

*Figure 1 continued*

animals per repeat collected from independent cohort). Similar results have been confirmed in one other independent trial (*Figure 1—figure supplement 1*).

DOI: https://doi.org/10.7554/eLife.50069.002

The following source data and figure supplement are available for figure 1:

**Source data 1.** Metabolomics data for D4 and D12 GRU101 and GRU102.

DOI: https://doi.org/10.7554/eLife.50069.004

**Figure supplement 1.** Metabolomics profile of GRU102 and its transgenic controls (GRU101) (independent repeat).

DOI: https://doi.org/10.7554/eLife.50069.003

Ornitine over total AA levels; for Succinyl CoA, we took the sum of Methionine and Valine over total AA levels. This analysis of the AA profile demonstrated that, in comparison to age-matched controls, the percentage of glucogenic AA forming forming alpha-Ketoglutarate (aKG) was significantly increased in old GRU102 animals (*Figure 1D*, *Figure 1—figure supplement 1*).

From the OA profiles, we found only an insignificant trend towards higher aKG levels in old GRU102 compared to GRU101 (*Figure 1E*, *Figure 1—figure supplement 1*). However, both of the TCA cycle intermediates, fumarate and malate, were significantly reduced in old GRU102 (*Figure 1E*, *Figure 1—figure supplement 1*). These observations suggest a specific disruption in the TCA cycle that affects some but not all of the TCA intermediates.

## Reduced alpha-Ketoglutarate dehydrogenase activity in GRU102

To help identify which reaction(s) in and around the TCA cycle were most strongly altered in GRU102, we next performed a computational metabolic flux balance analysis (FBA) using transcriptomics and metabolic data from young GRU101 and GRU102 animals as constraints. Modeling took into consideration gene expression data (RNAseq) and data related to nutritional uptake and metabolic rates, specifically pharyngeal pumping, oxygen consumption, and body size (*Supplementary file 1*). Gene expression data were used to curate *C. elegans* condition-specific metabolic networks by modifying a genome-scale metabolic model of *C. elegans* (iCEL 1273) (*Yilmaz and Walhout, 2016*) using the Gene Inactivity Moderated by Metabolism and Expression (GIMME) algorithm (*Becker and Palsson, 2008*). Specifically, we performed parsimonious flux balance analysis (pFBA; *Yilmaz and Walhout, 2016*) using the COnstraint-Based Reconstruction and Analysis (COBRA) toolbox (*Schellenberger et al., 2011*) to determine the metabolic flux distribution corresponding to maximum biomass production (growth and storage; *Yilmaz and Walhout, 2016*). We subsequently carried out flux variability analysis (FVA) to determine the range of flux values for individual reactions that maintains the same optimal biomass production for GRU101 and GRU102 (for more details on the flux analysis see Materials and methods), with the aim of identifying reactions and metabolites central to the metabolic alterations in GRU102.

Comparing the metabolic flux distributions between young GRU101 and GRU102, the FBA suggested a more than two-fold flux changes in several reactions related to aKG, including a ~ 3 fold reduced production of cytosolic aKG from glutamate and a ~ 10 fold increased production of cytosolic aKG from 3phosphohydroxypyruvate. The model predicted a reversal in the direction of flux in the alanine-pyruvate cycle, where, in young GRU101, alanine and aKG are converted to pyruvate and glutamate, but in GRU102, pyruvate and glutamate are converted to alanine and aKG (*Figure 2*). Scanning through all results of FVA, we identified eight reactions that exhibit such directional inconsistency, where the flux intervals did not have consistent signs (direction) between GRU101 and GRU102 (*Supplementary file 2*). Seven out of the eight reactions were involved in transport, exchange, and co-factor (ADP, NADH) production, while the remaining reaction participates in the alanine-pyruvate cycle. Taken together, the pFBA and FVA results motivated us to investigate alterations specifically in the metabolism of aKG as a potential factor involved in the metabolic defects of GRU102.

The reduction in pyruvate observed in GRU102 potentially suggests increased Lactate Dehydrogenase (LDH) activity. The accumulation of aKG, coupled with the reduction in fumarate and malate, is consistent with a reduction in the activity of aKG dehydrogenase (aKGDH) or with alterations in enzymes downstream of aKG. Among those downstream enzymes, Succinyl CoA Synthetase (SCS) is

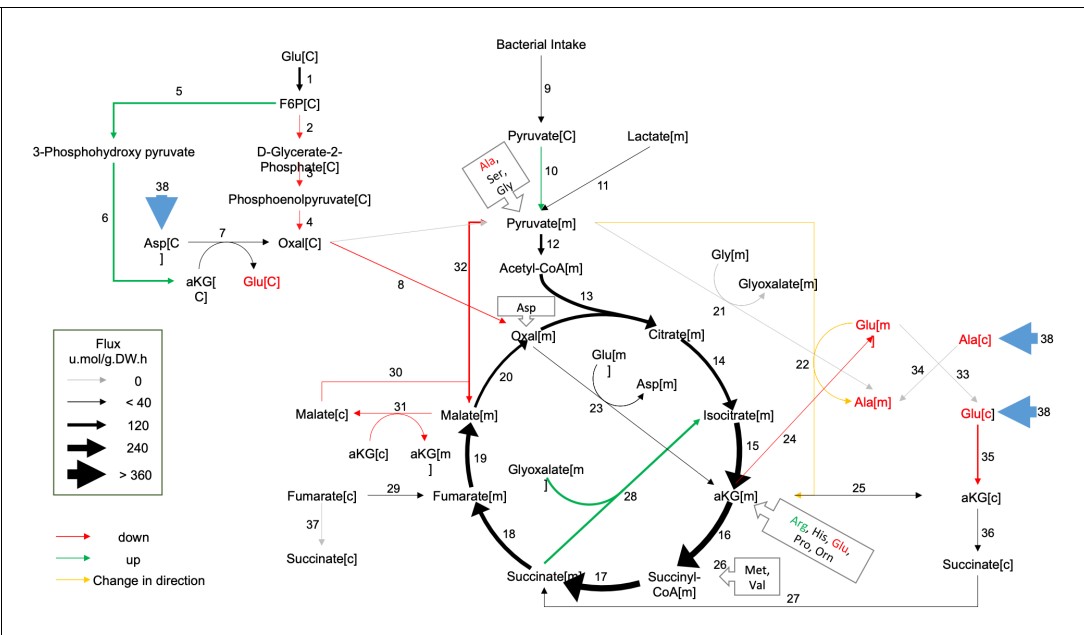

**Figure 2.** Metabolic flux in day 4 GRU102 modeled by metabolic flux balance analysis. Changes in flux are shown relative to GRU101 controls. Refer to Source data file for raw flux values.

DOI: https://doi.org/10.7554/eLife.50069.005

The following source data is available for figure 2:

**Source data 1.** Raw flux values derived from flux balance analysis.

DOI: https://doi.org/10.7554/eLife.50069.006

an important regulator of mitochondrial integrity including in neurons (*Zhao et al., 2017*). Increased levels of another downstream enzyme, Malate Dehydrogenase (MDH), have previously been found in the brain of AD patients (*Paterson et al., 2016*). We therefore directly examined the enzymatic activity of LDH, aKGDH, SCS and MDH. We evaluated the maximum rate for LDH, aKGDH, SCS and MDH in extracts of purified mitochondria derived from all cells of age-matched GRU102 or GRU101 controls. Out of these four enzymes, we found a significant change (reduction) only in the activity of aKGDH (*Figure 3B* for aKGDH, *Figure 3—figure supplement 1* for data on the other three enzymes). We observed a significant reduction in aKGDH activity relative to age-matched GRU101 controls in old but not young GRU102.

## aKGDH knockdown in controls recapitulated metabolic detriments of GRU102

To examine if the reduction in aKGDH activity alone was able to recapitulate the metabolic deficits observed in GRU102, we exposed GRU101 and GRU102 animals to RNAi against the *C. elegans* aKGDH gene (*ogdh-1*) and investigated their effects on metabolism using a Seahorse bioanalyzer. To avoid confounding developmental effects, we started aKGDH knockdown *via* RNAi treatment only from adulthood (day 3) in both backgrounds. We found that aKGDH knockdown significantly reduced both basal and maximal respiration in GRU101, an effect similar to the defects in GRU102 (*Figure 3C*). Notably, as for GRU102, the response to the addition of chemical uncoupler appeared both delayed and blunted in GRU101 with adult aKGDH knockdown. In fact, the oxygen consumption rate (OCR) profile of GRU101 with aKGDH knockdown closely mirrored that of GRU102 (*Figure 3C*). To evaluate if loss of aKGDH might be a compensatory mechanism, we also carried out lifespan studies in GRU102 subject to RNAi knockdown of aKGDH but found that further loss of this enzyme did not significantly affect lifespan, suggesting that reduced aKGDH activity is not protective in GRU102 (*Figure 3—figure supplement 2*). We also evaluated metabolic effects of aKGDH knockdown in GRU102 and found that loss of aKGDH similarly did not normalize but instead significantly

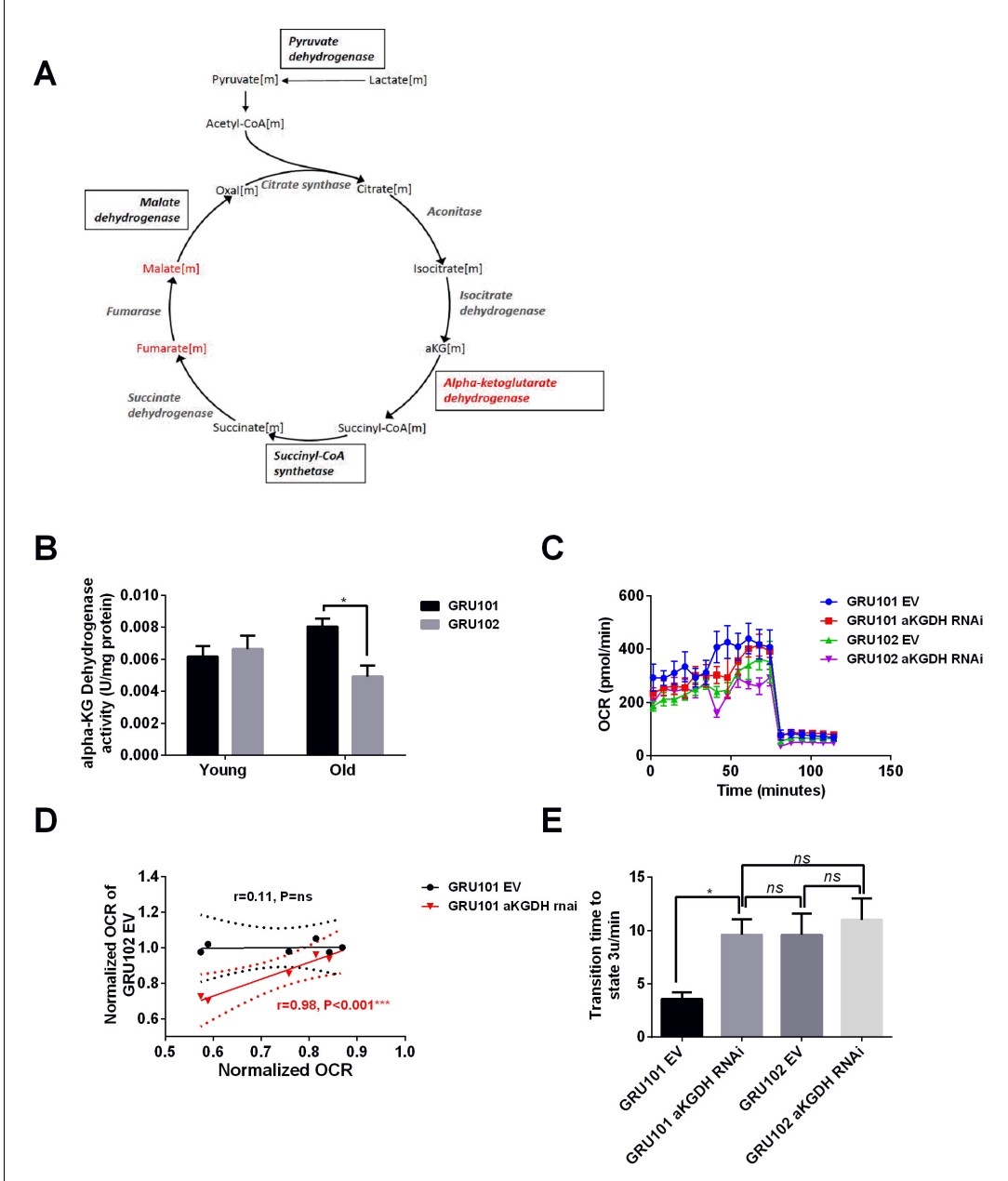

**Figure 3.** aKGDH knockdown recapitulated metabolic detriments of GRU102. (**A**) Diagram illustrating changes in key enzymes and metabolites involved in the TCA cycle. (**B**) alpha-KG dehydrogenase activity in GRU102 and GRU101 controls (Two-way ANOVA Dunnett's multiple comparisons test, p<0.05, *; n = 4–8 repeats per condition, with approximately 1500 animals collected from independent cohorts). (**C**) Oxygen consumption rate (OCR) profiles comparing GRU101 and GRU102 fed with aKGDH RNAi or EV (n = 6 repeats per group, with 10 animals per repeat). (**D**) Scatter plot illustrating high degree of correlation in OCR timecourse of GRU102 and GRU101 fed with aKGDH RNAi. OCR values were normalized to average uncoupled values of GRU101 (n = 6 measurements for each condition). (**E**) Time taken to reach fully uncoupled respiration (state 3 u) after addition of chemical uncoupler, defined as OCR rising to a value of two standard deviation (SD) above average coupled/basal OCR (n = 6 repeats per group, with 10 animals per repeat, One-way ANOVA Sidak's multiple comparison, p<0.05, *).

DOI: https://doi.org/10.7554/eLife.50069.007

The following source data and figure supplements are available for figure 3:

**Source data 1.** Raw data for enzymatic activity assays.
DOI: https://doi.org/10.7554/eLife.50069.010
**Source data 2.** Raw data for Seahorse oxygen consumption rate profile for aKGDH knockdown animals.
DOI: https://doi.org/10.7554/eLife.50069.011

*Figure 3 continued*

**Figure supplement 1.** TCA cycle enzyme activities.
DOI: https://doi.org/10.7554/eLife.50069.008
**Figure supplement 2.** Survival curve of GRU102 fed with empty vector (EV) or aKGDH RNAi.
DOI: https://doi.org/10.7554/eLife.50069.009

worsened the metabolic flux in GRU102 animals (*Figure 3C*), further confirming that reduction in aKGDH is detrimental in AD.

Comparing OCR profiles between GRU101 animals subject to aKGDH knockdown and GRU102, we noticed that their timeseries of OCR values following addition of the uncoupler appeared to be similar. To further assess how well the OCR profile of GRU101 fed with aKGDH RNAi correlates with GRU102, we plotted a scatter diagram of OCR of GRU102 vs GRU101 fed with aKGDH RNAi or EV just before and during uncoupling (time measurement 6–12 of *Figure 3C*, after addition of FCCP uncoupler). The correlation coefficient, r, can be used to evaluate how well the respective conditions correlate with each other, with r value close to one being indicating a high degree of correlation. We found that the OCR during uncoupling of GRU102 correlates well with that of GRU101 fed with aKGDH RNAi (r = 0.98, p<0.001) but not for animals fed EV (r = 0.33, p=ns, *Figure 3D*), illustrating that aKGDH knockdown causes GRU101 to experience deficits in the response to uncoupling that resembles GRU102. One way in which the OCR time course of GRU101 with aKGDH knockdown resembles GRU102 is an increase in the time taken to reach maximal uncoupled respiration (state 3 u), following addition of uncoupler (FCCP). To directly quantify this effect, we compared the time needed for OCR values to rise to at least two standard deviation above the average coupled (basal) respiration, finding that, while GRU101 rapidly responded to addition of FCCP, both GRU102 and GRU101 subject to aKGDH knockdown suffer a significant delay (*Figure 3E*). Interestingly, aKGDH knockdown did not further delay uncoupled respiration in GRU102. Together, these results suggest that aKGDH knockdown is sufficient to recapitulate key metabolic deficits seen in GRU102.

## Oxidative stress as a potential cause of mitochondrial dysfunction in the GRU102 animals

aKGDH has been reported to be sensitive to inactivation by reactive oxygen species (ROS), in part through the oxidation of lipoic acid bound covalently to the E2 subunit of aKGDH (*Tretter and Adam-Vizi, 2005*). Oxidative stress is thought to contribute to the pathophysiology of AD (*Swomley and Butterfield, 2015*; *Tramutola et al., 2017*) and Aβ has been reported to directly promote oxidative stress (*Mattson and Mattson, 2002*; *Huang et al., 1999*). This suggests that ROS-related oxidative damage might be a plausible cause of mitochondrial dysfunction, including aKGDH inactivation.

Protein carbonylation is a consequence of oxidative damage to protein and protein carbonyl content (PCC) is routinely used to assess ROS-related protein oxidation (*Suzuki et al., 2010*). To test if oxidative damage to protein might be a plausible cause for the enzymatic and metabolic defects in GRU102, we next determined PCC in whole lysate and in mitochondrial fractions of age-matched GRU102 and GRU101 controls. While total PCC in whole body worm lysate was unchanged in GRU102 (*Figure 4A*), we found a significant elevation in mitochondrial protein carbonyls (mtPCC) even in young GRU102 when compared to GRU101 (*Figure 4B*). These data suggested that ROS-mediated damage, specifically within the mitochondrial compartment, is an early pathogenic event occurring in GRU102.

To further examine whether the metabolic detriments and ROS-related damages were mediated by cell-autonomous mechanisms and primarily confined in the neurons, we performed whole body mitoSOX and mitoTracker staining of the animals, to respectively examine ROS localization and mitochondrial morphology in neuronal versus non-neuronal compartments of the animals. However, we found that the mitoSOX dye predominantly localizes at the pharyngeal bulb as also previously reported (*Dingley et al., 2010*), and the resolution we were able to get was not high enough to clearly differentiate between neuronal and other cellular structure (*Figure 4—figure supplement 1*). Similarly, we also could not distinguish between neuronal *vs* other cellular compartment from the mitotracker stain (*Figure 4—figure supplement 1*).

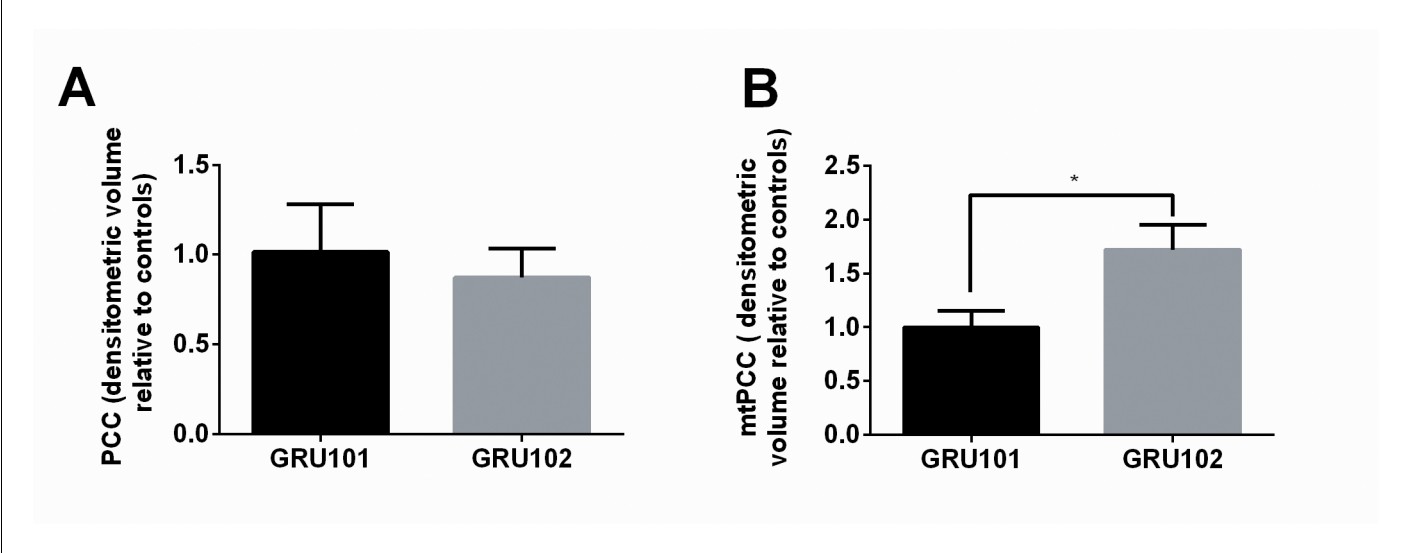

**Figure 4.** Evidence for ROS-related oxidative protein damage in GRU102. (**A**) Densitometric analysis of protein carbonyl content (PCC) from whole lysate in young GRU101 and GRU102. (**B**) Densitometric analysis of mitochondrial protein carbonyl content (mtPCC) from whole lysate in young GRU101 and GRU102 (unpaired t-test, p<0.05, *; n = 3–7 repeats per condition, with approximately 1500 animals per repeat collected from independent cohorts).

DOI: https://doi.org/10.7554/eLife.50069.012

The following source data and figure supplement are available for figure 4:

**Source data 1.** Protein carbonyl content (PCC) in D4 GRU101 and GRU102.

DOI: https://doi.org/10.7554/eLife.50069.014

**Figure supplement 1.** Mitosox and mitotracker staining of GRU101 and GRU102 animals.

DOI: https://doi.org/10.7554/eLife.50069.013

## Metformin improved Aβ pathology in GRU102

Given evidence for early disruption in energy metabolism and ROS-mediated damage in GRU102, we next examined whether Metformin, an anti-diabetic drug and metabolic modulator that has been shown previously to improve stress resistance and increase TCA cycle metabolism in Wildtype (WT) *C. elegans* (*De Haes et al., 2014*; *Onken and Driscoll, 2010*), was able to rescue Aβ-induced pathology in GRU102. We found that Metformin at a dose of 50 mM rescued the lifespan deficit of GRU102 by normalizing its median lifespan to that of untreated GRU101 (*Figure 5A*, *Supplementary file 3*). Metformin treatment also strengthened stress defense of GRU102, as seen in the reduced in percentage of dead GRU102 following challenge with Paraquat (*Figure 5B*), a herbicide that induces superoxide production (*Wiemer and Osiewacz, 2014*). While old GRU102 exhibited significantly impaired respiratory capacity, Metformin treatment significantly increased maximal respiratory capacity of GRU102 to a level that was, however, still lower than that of untreated GRU101 (*Figure 5C*).

To determine which metabolic substrates were involved in this rescue by Metformin, we again performed targeted metabolomic analysis and compared AA, AC and OA intermediates between Metformin treated and untreated old GRU102 (day 12). We found that Metformin treatment increased the abundance of many AA in GRU102, with Alanine, Glycine, Proline and Serine consistently increased in both replicates (*Figure 5D*, *Figure 5—figure supplement 1*). Interestingly, at the same time that it increased AA related metabolites, Metformin treatment significantly reduced the level of fat-derived C18:1-carnitine (*Figure 5E*), suggesting that Metformin reduced the amount of energy derived from lipid metabolism relative to that derived from AA metabolism. Metformin treatment also reduced the level of lactate in GRU102 while at the same time increasing the levels of pyruvate, fumarate and malate. Levels of malate were in fact significantly higher than those in GRU101 (*Figure 5F*). Our metabolomics findings were consistent with a previous proteomic analysis of Metformin-treated WT *C. elegans*, which suggested an enrichment of proteins involved in

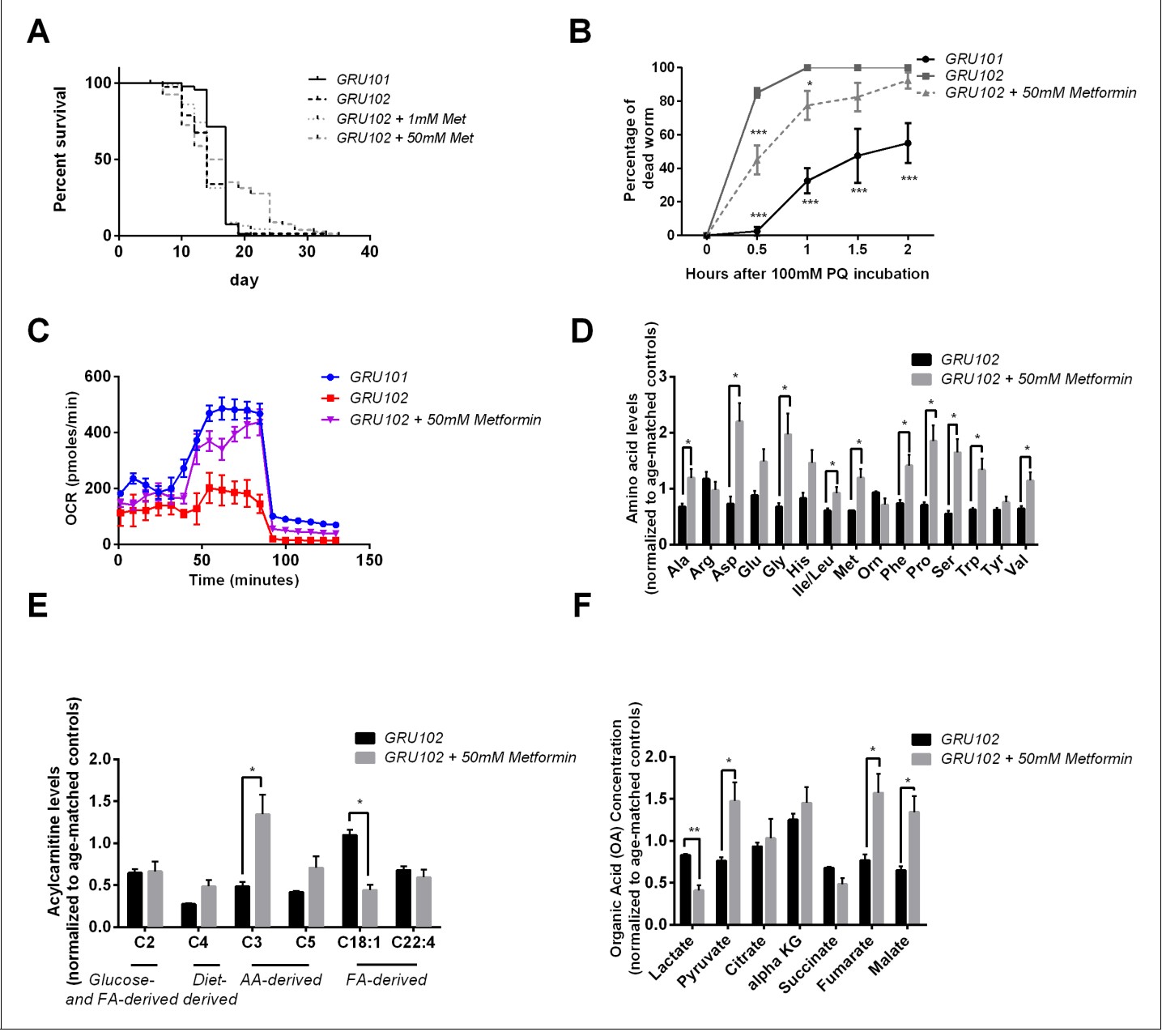

**Figure 5.** Effects of Metformin on Aβ-induced pathology in GRU102. (**A**) Survival analysis of GRU102 animals treated with Metformin. At a dose of 50 mM, Metformin treatment significantly improved the lifespan of GRU102 (log-rank test p<0.005; n = 70–100 animals per condition). Similar result has been confirmed in two other independent trials (**Supplementary file 3**). (**B**) Time course of the percentage of dead animals observed as a function of time following treatment with 100 mM Paraquat (PQ). Metformin treatment improved PQ-stress resistance of GRU102 (Two-way ANOVA Dunnett's multiple comparison, p<0.05, *; p<0.001, ***; n = 4 repeats per condition, with 10–20 animals per repeat). Similar result has been confirmed in one other independent trial. (**C**) Metabolic flux profile of Metformin-treated GRU102 (n = 6 repeats per condition, with 10 animals per repeat). Similar result has been confirmed in two other independent trials. (**D**) Amino acid, (**E**) Acylcarnitine, (**F**) Organic acid profile of Metformin-treated GRU102 on day 12. All values were normalized to respective protein level and then to age-matched GRU101 control (unpaired t-test, p<0.05, *; p<0.005, **; n = 3 repeats per condition with approximately 3000 animals per repeat collected in three independent cohorts). Similar result has been confirmed in one other independent trial (**Figure 5—figure supplement 1**).

DOI: https://doi.org/10.7554/eLife.50069.015

The following source data and figure supplements are available for figure 5:

**Source data 1.** Metabolomics data for D12 GRU102 and D12 GRU102 treated with Metformin.
DOI: https://doi.org/10.7554/eLife.50069.020

**Source data 2.** Raw data for Seahorse oxygen consumption rate profile for D12 Metformin-treated GRU102.

*Figure 5 continued on next page*

*Figure 5 continued*

DOI: https://doi.org/10.7554/eLife.50069.021

**Figure supplement 1.** Effects of Metformin on Aβ-induced pathology in GRU102 (Independent repeat).

DOI: https://doi.org/10.7554/eLife.50069.016

**Figure supplement 2.** Metabolomics analysis of Metformin treatment.

DOI: https://doi.org/10.7554/eLife.50069.017

**Figure supplement 3.** Levels of cluster one metabolites in old (day 12) animals.

DOI: https://doi.org/10.7554/eLife.50069.018

**Figure supplement 4.** Sulfamethoxazole (SMX) did not rescue metabolic flux or lifespan detriments in GRU102.

DOI: https://doi.org/10.7554/eLife.50069.019

branched-chain amino acid degradation and TCA cycle metabolism (*De Haes et al., 2014*). Collectively, these data suggest that Metformin improves energy substrate availability, and that such metabolic improvement is helpful against Aβ-mediated mitochondrial toxicity in GRU102.

We next compared the metabolic state of GRU101 and GRU102 animals at different ages (day 4 and 12), and compared these with 12 day old Metformin-treated GRU102 (*Figure 5—figure supplement 2*). The Heatmap of this analysis revealed an interesting cluster of metabolites (Cluster 1: Ornitine, aKG, Pyruvate, Histidine, Alanine and Glutamine) that showed a strong age-dependent decline that could be rescued by Metformin treatment. To examine how ages, genotypes and metformin treatment, respectively, affect the metabolites in this cluster, we performed a Principal Component Analysis (PCA) and also individually compared the metabolites of cluster 1. PCA analysis showed that genotype only had a significant influence on the metabolites in young but not old animals, where levels were similarly reduced in both GRU101 and GRU102 (*Figure 5—figure supplement 2*, *Figure 5—figure supplement 3*). Individually, levels of Ornitine, aKG, Alanine and Glutamine significantly reduced in young GRU102 compared to young GRU101 (*Figure 5—figure supplement 2C*). Age completely separated the metabolites for GRU101 (*Figure 5—figure supplement 2B*) while only Pyruvate, Histidine and Alanine significantly declined with age in GRU102 (*Figure 5—figure supplement 2D*). Interestingly, Metformin treatment reverted the levels of these metabolites (Pyruvate, Histidine, Alanine and Glutamine) in old GRU102 back to that of the young GRU102 (*Figure 5—figure supplement 2C*), suggesting that Metformin reverses the aging-related metabolic decline of metabolites in cluster 1.

Metformin has been previously shown to impact lifespan in *C. elegans* through inhibition of folate production by the microbiome. Sulfomethoxazole (SMX), a sulfonamide antibiotics that blocks folate synthesis, has similarly been shown to extend lifespan in nematodes, establishing a link between folate inhibition and lifespan extension (*Virk et al., 2012*). To determine if the beneficial effects of Metformin in GRU102 were mediated by microbiome folate inhibition, we investigated whether addition of SMX phenocopies the beneficial effects of Metformin in GRU102. We found that SMX did not rescue the metabolic flux impairment or the lifespan detriment of GRU102 (*Figure 5—figure supplement 4*). These observations suggest that the beneficial effects of Metformin in GRU102 were unlikely to be mediated predominantly by inhibition of microbiome folate production.

## General protein aggregation as a secondary event to metabolic alterations in AD

Increased oxidative stress can disrupt protein homeostasis (proteostasis) through irreversible oxidation and subsequent modifications of proteins, and such damage also impairs protein degradation (*Squier, 2001*). Proteostasis is energetically expensive, because protein degradation, synthesis and quality control, required for turnover of misfolded proteins, are all energy-intensive processes (*Hartl et al., 2011*). Given that we observed increased mtPCC levels and deficits in bioenergetics (ATP) (*Fong et al., 2016*), we next asked whether global proteostasis was affected in the GRU102 animals. A general disruption of proteostasis is different from specific formation of Aβ-derived aggregates in that it is expected to affect a wide range of endogenous, aggregation-prone peptides in addition to Aβ itself.

To determine whether general protein aggregation occurs in the Aβ-expressing animals, we examined the levels of insoluble proteins *via* SDS-PAGE (*David et al., 2010*). While this method provides a general indication on the levels of insoluble proteins, we noticed a degree of variability in

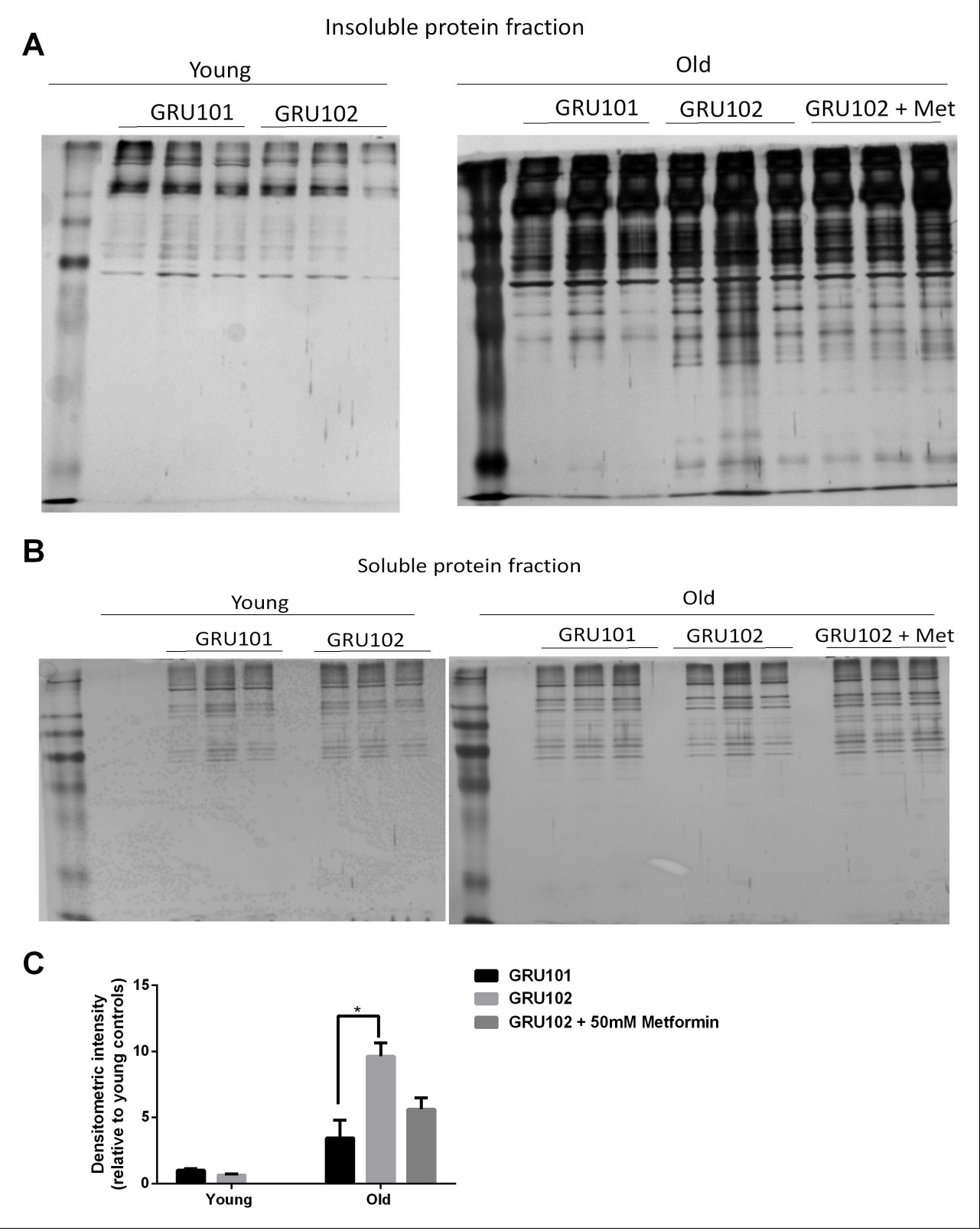

**Figure 6.** Levels of insoluble protein in GRU102. (**A**) Silver stain of SDS-Page gel of detergent-insoluble protein, loaded based on five times the volume equivalent of 2 ug soluble protein determined by Bradford protein assay. (**B**) Silver stain of SDS-Page gel of 0.2 ug detergent-soluble protein determined by Bradford protein assay. (**C**) Normalized densitometric intensity corrected for differences in protein loading evaluated from soluble protein gel (Densitometric intensity of Gel A/Densitometric intensity of Gel B; Two-way ANOVA Dunnett's multiple comparisons test, $p < 0.05$, n = 3

*Figure 6 continued on next page*

*Figure 6 continued*

repeats for all conditions, each repeat contains approximately 1500 animals collected in independent cohort). Similar result was observed in another independent repeat (*Figure 6—figure supplement 1*).

DOI: https://doi.org/10.7554/eLife.50069.022

The following source data and figure supplement are available for figure 6:

**Source data 1.** Protein aggregate levels determined by insoluble gel.

DOI: https://doi.org/10.7554/eLife.50069.024

**Figure supplement 1.** Levels of insoluble protein in GRU102 (independent repeat).

DOI: https://doi.org/10.7554/eLife.50069.023

the intensity of insoluble gel among samples within the same condition (*Figure 6A*). To normalize for potential differences in protein loading, we ran a separate SDS-PAGE of the soluble protein fractions (*Figure 6B*) and normalized the densitometric intensity of the insoluble gel to that of the soluble gel (*Figure 6C*). Using this normalization we found that, when the animals were young, no difference in the levels of insoluble protein was detected between GRU102 and GRU101 (*Figure 6C*), indicating that the proteostasis machinery of young GRU102 is still relatively intact. However, aged GRU102 displayed significantly higher levels of insoluble protein compared to age-matched GRU101 (*Figure 6C*).

As discussed above, treatment with Metformin results in substantial rescue of metabolic defects in GRU102 animals (*Figure 5*). Because metabolic alterations are known to affect proteostasis (*Ruan et al., 2017*; *Patel et al., 2017*; *Fang et al., 2019*; *Guerrero-Gómez et al., 2019*), we next investigated whether metabolic rescue following treatment with Metformin also normalizes levels of global protein aggregation in aged GRU102. Even though we observed a trend towards lower levels of insoluble protein in the Metformin-treated GRU102, the difference did not reach statistical significance (*Figure 6C*), possibility due to the variability of the gel. However, we also observed that the levels of protein aggregates in the Metformin-treated GRU102 appear similar to the GRU101 and the difference between GRU101 and Metformin-treated GRU102 was no longer statistically significant (*Figure 6C*, One-way ANOVA multiple comparisons test p=ns). Metformin treatment therefore appears to benefit GRU102 in terms of proteostasis but it did not fully rescue the increase in protein aggregates.

## Discussion

Our study employed GRU102, a strain constitutively expressing low-levels of human Aβ1–42 pan-neuronally. We have previously reported that energy deficits and reduced activities of mitochondrial ETC complexes are early events in GRU102, well before any detectable Aβ aggregates are present (*Fong et al., 2016*). Here, we have shown that key energy substrates, in particular glucogenic AA that form pyruvate and aKG, as well as the TCA cycle intermediates fumarate and malate, are selectively affected in GRU102. Metabolic flux balance analysis, incorporating transcriptomics and metabolic data (oxygen intake and bacterial consumption), suggested alterations in transport or metabolism of aKG to be involved in these defects. Indeed, enzymatic assays of key TCA cycle enzymes confirmed deficiency in a rate-limiting TCA cycle enzyme, aKGDH, involved in aKG metabolism.

Given that aKGDH is a rate-limiting enzyme in the TCA cycle (*Huang et al., 2003*), reduced aKGDH activity is a plausible explanation for reduced flux through the TCA cycle, ATP production and maximum respiratory capacity. Inhibition of aKGDH activity, as observed in GRU102, may therefore partially explain key metabolic defects observed in GRU102. Inactivation of aKGDH may contribute not only to the observed reduction in levels of TCA cycle metabolites downstream of aKG (malate and fumarate), but also explain the general reduction in spare respiratory capacity and energy deficits in the GRU102 animals (*Butterfield et al., 2006*). To test this hypothesis, we carried out aKGDH RNAi knockdown in adult GRU101 as well as GRU102 animals and found GRU101 fed on aKGDH RNAi suffered metabolic detriments that closely resembles GRU102. Exposure to aKGDH RNAi of GRU102, which are already experiencing reduced aKGDH activity, similarly did not improve their metabolic defects and instead resulted in further detriments. These results suggest that reduction in aKGDH activity is detrimental to metabolic capacity in adults and that it alone can

recapitulate the metabolic deficits of AD in *C. elegans*. This, of course, does not mean that inactivation of aKGDH is the single cause of the metabolic detriments in Aβ1–42-expressing *C. elegans*, but it suggests that inactivation of specific enzymes involved in the TCA may be a key mechanism of Aβ toxicity.

Our study revealed mitochondrial-specific ROS as a potential early cause of mitochondrial dysfunction in our AD model, as shown by the early (day 4) increase in mtPCC but not in global PCC. In vitro experiments have shown that the mitochondrial enzyme aKGDH is particularly sensitive to inactivation by ROS (hydrogen peroxide) (*Tretter and Adam-Vizi, 2000*), consistent with the involvement of this enzyme in GRU102 metabolic defects. Site-specific oxidative modification to mitochondrial but not global protein is one possible mechanism to explain the reduction of aKGDH activity observed in old GRU102. These findings are consistent with other reports showing that mitochondrial enzymes involved in energy regulation are known to be redox sensitive and often downregulated in AD (*Butterfield and Halliwell, 2019*; *Mastrogiacomo et al., 1996*).

Remarkably, we found that metabolic deficits and increased mtPCC were detectable in young GRU102, while increased global protein aggregation was only seen in aged GRU102. This sequence of events suggests that impairment of proteostasis may be late or secondary to metabolic dysfunction and/or to oxidative inactivation of TCA cycle enzymes (*Figure 7*).

It is intriguing that GRU102 animals displayed such broad metabolic changes, given that the Aβ transgene is only expressed in neurons. This raises the question of whether these metabolic effects are primarily confined to the neurons, or if non-cell-autonomous effects contribute to these phenotypes. While our data currently do not address this question, the Aβ-induced oxidative stress hypothesis of AD (see Introduction paragraph 2) supports oxidative stress as a direct and cell-autonomous mechanism of Aβ-induced toxicity (*Butterfield and Halliwell, 2019*). The confined effects of oxidative stress in Aβ-expressing neurons is also supported by clinical findings that ROS-mediated elevation in PCC are only observed in brain regions of AD subjects rich in Aβ-plaques, but not in other brain regions devoid of Aβ-plaques (*Hensley et al., 1995*).

On the other hand, existing literature has suggested the activation of stress response mechanism and proteotoxicity as non-cell-autonomous consequences in protein-misfolding diseases including AD (*Nussbaum-Krammer and Morimoto, 2014*), as activation of stress response, such as the heat shock response or unfolded protein response, functions at a whole animal level, beyond the immediate affected cell (*Nussbaum-Krammer and Morimoto, 2014*). Studies in *C. elegans* have also shown that induction of mitochondrial stress in neurons can lead to non-cell-autonomous mitochondrial stress response in non-neuronal compartment such as intestine (*Durieux et al., 2011*). The mechanism of the impairment in mitochondrial health observed in GRU102 may therefore not be confined to neuronal cells only. The pan-neuronal unc-119 promoter that we use to drive Aβ expression in GRU102 has been suggested to have low expression in intestine, and such expression might lead to detriments in that compartment not confined in the neurons.

Metformin has previously been shown to strengthen stress defense, and to increase branched-chain AA degradation and TCA metabolism in WT *C. elegans* (*De Haes et al., 2014*; *Onken and Driscoll, 2010*). We found that Metformin treatment normalizes lifespan and mitochondrial function in GRU102 up to the level of untreated GRU101 controls. Lifespan of the Metformin-treated GRU102 was even longer than the untreated GRU101. Consistent with previous studies showing that Metformin improves AA and TCA cycle metabolism in WT *C. elegans* (*De Haes et al., 2014*), we demonstrated that Metformin treatment also restores respiratory capacity and improves the availability of glucogenic AA and TCA cycle intermediates in the GRU102 animals.

We further found that Metformin-treated GRU102 showed a trend towards lower levels of protein aggregates. However, due to the variability inherent in this method, the reduction was not statistically significant, and we could not conclusively show that Metformin reduces protein aggregation in GRU102. However, protein aggregates levels in Metformin-treated GRU102 were also no longer significantly different from GRU101, suggesting that Metformin treatment had some benefit on this parameter. While more work needs to be carried out to confirm the causal relationship between metabolic dysfunction and protein aggregation, our observation is consistent with findings that metabolic health and proteostasis are closely related to each other (*Ruan et al., 2017*; *Patel et al., 2017*; *Fang et al., 2019*; *Guerrero-Gómez et al., 2019*).

Given that Metformin can penetrate through the blood brain barrier (*Ying et al., 2015*), our findings suggest Metformin as an alternative therapy for AD. Clinical trials of Metformin in MCI and mild

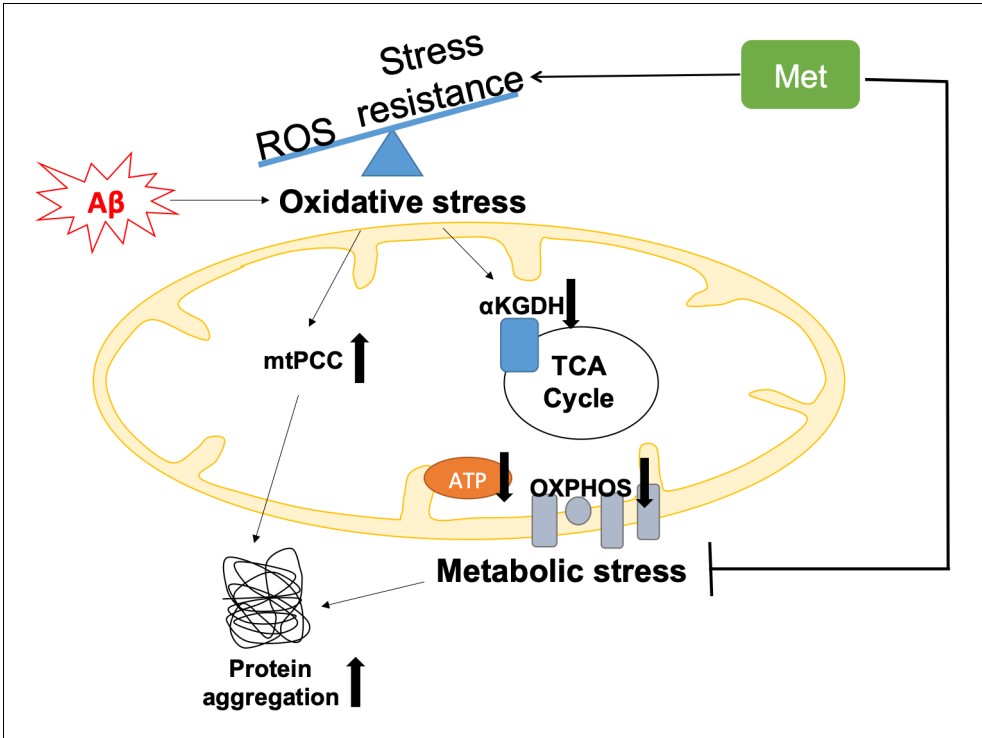

**Figure 7.** Graphical summary of the model of Aβ pathology in GRU102. Pan-neuronal Aβ expression leads to an imbalance between reactive oxygen species (ROS) production and antioxidant (AOX) defense, resulting in oxidative stress as an early event in GRU102. Increase in mitochondrial protein carbonyl content (mtPCC) was also observed as an early phenomenon in GRU102. Elevated oxidative stress further affects ROS-sensitive Tricarboxylic Acid (TCA) cycle enzyme, including alpha-ketoglutarate dehydrogenase (aKGDH), resulting in a reduction of its activity as well as altered levels of TCA metabolites (aKGDH, fumarate and malate) in old GRU102. Together with the reduction in electron transfer chain (ETC) complex I and IV activities as well as low ATP levels in the GRU102 animals we reported previously (*Fong et al., 2016*), these phenomena result in metabolic stress. Proteostasis, an energy-intensive process, is impaired by metabolic and oxidative stress, resulting in an increase in general protein aggregates. Treatment with Metformin (Met) increases stress resistance and rescues metabolic defects of GRU102. Metformin-treated GRU102 also appears to reduce the increase in protein aggregates in GRU102, even though it could not fully rescue this phenotype.

DOI: https://doi.org/10.7554/eLife.50069.025

AD patients have also suggested improvement in cognitive function in the Metformin-treated groups (*Koenig et al., 2017*; *Luchsinger et al., 2016*). However, we noted that the dose of Metformin used in this study (50 mM, equivalent to 8.25 g/L) is approximately five times higher than the therapeutic concentrations of plasma Metformin reported in humans (up to 1.8 g/L) (*Kajbaf et al., 2016*). This difference could be due to the low permeability of *C. elegans* cuticle to exogenous drug compounds, which often leads to the much higher drug concentration needed for therapeutic effects (*Chisholm and Xu, 2012*).

In conclusion, our results suggest metabolic failure as an early and central event in AD pathogenesis. Metabolic interventions that normalize metabolic dysfunction may therefore be beneficial to prevent or reduce AD pathogenesis.

# Materials and methods

**Key resources table**

| Reagent type (species) or resource | Designation | Source or reference | Identifiers | Additional information |
|---|---|---|---|---|
| Strain (*C. elegans*) | GRU101 gnals1 [myo-2p::yfp] | Caenorhabditis Genetics Center | | |
| Strain (*C. elegans*) | GRU102 Gnals2 [myo-2p::YFP + unc-119p::Abeta1-42] | Caenorhabditis Genetics Center | | |
| Commercial assay or kit | Qiagen RNAeasy micro kit | Qiagen | Cat. No 74004 | |
| Commercial assay or kit | Alpha-Ketoglutarate Dehydrogenase Activity Colorimetric A ssay Kit | BioVision Inc | Cat. No K678 | |
| Commercial assay or kit | Succinyl-CoA Synthetase Activity Colorimetric Assay Kit | BioVision Inc | Cat. No K597 | |
| Commercial assay or kit | Malate Dehydrogenase Activity Colorimetric Assay Kit | BioVision Inc | Cat. No K654 | |
| Commercial assay or kit | Lactate Dehydrogenase Activity Colorimetric Assay Kit | BioVision Inc | Cat. No K726 | |
| Chemical compound, drug | 1,1-Dimethylbiguanide hydrochloride (Metformin) | Sigma Aldrich | Product no D150959 | |

## Nematode maintenance and drug/RNAi treatments

The transgenic strains, GRU101[myo-2p::yfp] and GRU102[myo-2p::yfp + unc-119p::Aβ1–42], previously reported in *Fong et al. (2016)* were used in all experiments. These animals were raised at 20℃ throughout the experiment and transferred to nematode growth medium (NGM) plate supplemented with 250 uM 5-fluoro-2′-deoxyuridine (FUDR; Sigma-Aldrich, St. Louis, USA) to prevent progeny production. Age-synchronized animals were obtained for all experiments via hypochlorite bleaching. Young animals refer to day four post-bleaching while old animals refer to day 12 post-bleaching. For Metformin treatment, Metformin Hydrochloride (Sigma-Aldrich, St Louis, United States) was dissolved in water and added into NGM agar to a final concentration of 50 mM. Animals were transferred to Metformin-containing plate on day three post-hatching at L4 stage. For RNAi treatment, RNAi feeding clones were obtained from the *C. elegans* ORF-RNAi Library (Vidal) from Source BioScience. L4 animals were fed with HT115(DE3) bacteria expressing either the empty vector (EV) or RNAi clone T22B11.5 (for aKGDH knockdown), on NGM plates containing 2 mM IPTG and 100 ug/ml Carbenicillin.

## RNA extraction

Freshly prepared bacteria (OP50 *E. coli*) were spotted on 94 cm NGM agar plates on the previous evening and allowed to dry. Synchronized, young adult worms were transferred to fresh plates. For maintaining synchronized populations, FUdR was added to NGMmedia. After two days of treatment with drug or vehicle, adult worms were washed off the plates into 15 ml tubes and then washed several times with fresh buffer until a clear solution was obtained. Clean worm pellets were then frozen and later used for RNA extraction. Total RNA was isolated using Qiagen RNAeasy micro kit (Qiagen, Hilden, Germany) following the standard protocol.

## RNA-Seq data analysis

Following RNA extraction, RNA was quantified photometrically with NanoDrop 2000 and stored at −80C until use. The integrity of total RNA was measured by Agilent Bioanalyzer 2100. For library preparation, an amount of 2 mg of total RNA per sample was processed using Illumina's RNA

Sample Prep Kit following the manufacturer's instruction (Illumina, San Diego, CA, USA). All single drugs were processed together and multiplexed onto one lane to minimize the batch effect. The different drug combinations were processed into two multiplexed lanes. In total, we ran three lanes in parallel with 20 samples in each and we had untreated N2 controls in each lane. Libraries were sequenced using Illumina HiSeq4000 sequencing platform (Illumina, San Diego, CA, USA) in a paired-end read approach at a read length of 150 nucleotides. Sequence data were extracted in FastQ format. The RNAseq reads from each sample were mapped to the reference *C. elegans* transcriptome (WBcel235) with kallisto (v0.44.0) (*Bray et al., 2016*) and sequence-based bias correction. The estimated counts were imported from kallisto to the R environment (v3.5.1) and summarized to gene-level using the tximport package (v1.6.0) (*Soneson et al., 2015*). The DESeq2 package (v1.30.0) was used to identify differentially expressed genes (DEGs) in all our analysis for a significance threshold $\alpha$ of 0.05 after correcting for multiple hypothesis testing through Independent Hypothesis Weighting (IHW package v1.6.0) (*Ignatiadis et al., 2016*).

## Sample extraction and metabolic profiling

Nematodes were collected in M9 buffer with two washes to remove excess bacteria. Nematode pellet was obtained by centrifugation at 4°C, snap-frozen in liquid nitrogen and stored in −80°C until homogenization was performed. Nematode pellet was homogenized in 50% acetonitrile, 0.3% formic acid. Methods of extraction and metabolic profiling for amino acid (AA), acylcarnitines (AC) and organic acids (OA) are performed as described in *Newgard et al. (2009)*, *Sinha et al. (2014)*, and *Muoio et al. (2012)*.

For AC and AA extraction, 100 μL of homogenate was extracted using methanol. The AC and AA extracts were derivatised with 3M Hydrochloric acid in methanol or butanol (Sigma Aldrich, USA) respectively, dried and reconstituted in methanol for analysis in LC-MS. For OA extraction, 300 μL of homogenate was extracted with ethylacetate, dried and derivatized with N,O-Bis(trimethylsilyl)trifluoroacetamide, with protection of the alpha keto groups using ethoxyamine (Sigma Aldrich, USA).

AC measurements were made using flow injection tandem mass spectrometry on the Agilent 6430 Triple Quadrupole LC/MS system (Agilent Technologies, CA, USA). The sample analysis was carried out at 0.4 ml/min of 80/20 Methanol/water as mobile phase, and injection of 2 μL of sample. Data acquisition and analysis were performed on Agilent MassHunter Workstation B.06.00 Software.

Methods of AA analysis were modified from *Newgard et al. (2009)* and *Sinha et al. (2014)*. Briefly, AA were separated using a C8 column (Rapid Resolution HT, 4.5 × 50 mm, 1.8 μm, Zorbax SB-C8) on a Agilent 1290 Infinity LC system (Agilent Technologies, CA, USA) coupled with quadrupole-ion trap mass spectrometer (QTRAP 5500, AB Sciex, DC, USA). Mobile phase A (10/90 Water/Acetonitrile) and Mobile phase B (90/10 Water/Acetonitrile) both containing 10 mM of Ammonium formate were used for chromatography separation. The LC run was performed at a flow rate of 0.6 mL min$^{-1}$ with initial gradient of 20%B for 0.5 min, then ramped to 100% B in 2.5 min, maintained for 0.5 min, followed by re-equilibrated the column to the initial run condition (20% B) for 2 min. All compounds were ionized in positive mode using electrospray ionization. The chromatograms were integrated using MultiQuant 3.0 software (AB Sciex, DC, USA).

Trimethylsilyl derivatives of OA were separated on a gas chromatography column (VF-1ms; 30 m x 0.25 mm x 1 μm) by an Agilent Technologies HP 7890A and quantified by selected ion monitoring on a 5975C mass spectrometer using stable isotope dilution. The initial GC oven temperature was set at 70°C, and ramped to 300°C at a rate of 40 °C/min, and held for 2 min.

The mixOmics (6.6.0) R package (*Rohart et al., 2017*) was used to create the heatmap (*Figure 5— figure supplement 2*) using the principal components (PCA) loadings, after scaling and centering, of the raw abundances of each metabolite.

## Metabolic Flux Balance Analysis

We employed parsimonious flux balance analysis (pFBA) to evaluate the metabolic reaction fluxes in young GRU101 and GRU102 animals. In pFBA (*Yilmaz and Walhout, 2016*), metabolic fluxes of each metabolites are computed based on the mass balance around every metabolite in the metabolic network under the following assumptions: (1) that the metabolism is operating at steady state (i.e. no change in the metabolite concentrations over time), (2) that the organism/cell optimizes its metabolism according to a biological objective, and (3) that the total absolute flux in the metabolic

network is minimized (*Lewis et al., 2010*). We performed pFBA using built in functions in the COBRA toolbox (*Schellenberger et al., 2011*) in MATLAB. Given the life history of *C. elegans*, we chose the maximization of biomass production for growth and storage (*Yilmaz and Walhout, 2016*).

We curated condition-specific metabolic network models for the GRU101 and GRU102 animals by pruning the *C. elegans* genome-scale metabolic model iCEL 1273 (*Yilmaz and Walhout, 2016*) based on strain-specific transcriptomic data, using the Gene Inactivity Moderated by Metabolism and Expression (GIMME) (*Becker and Palsson, 2008*) algorithm. Briefly, starting from the genome-scale metabolic network (iCEL 1273) (*Yilmaz and Walhout, 2016*), the algorithm removes reactions whose mRNA transcript levels are below a pre-determined threshold. We set the threshold at the 5[th] percentile of the normalized transcript level for each gene. The algorithm then uses flux balance analysis (FBA) to ensure that the reduced model is able to achieve the same optimal biological objective value as the full model. In the event that the resulting model is unable to achieve the same biological objective, GIMME determines the smallest set of previously removed reactions to be added back into the reduced model to replicate the optimal biological objective value of the full model. We confirmed that changing the threshold in the range between 1 and 50 percentile does not significantly affect the results of our analysis.

Using the condition-specific metabolic models for GRU101 and GRU102 animals, we then applied pFBA to determine metabolic fluxes based on the maximization of biomass production. We started by analyzing the metabolic fluxes of the controls (GRU101), using experimentally observed bacterial consumption rate as a constraint on the bacterial intake rate (*Yilmaz and Walhout, 2016*; *Smith, 1992*; *Van Voorhies, 2002*). We used the results of the pFBA of GRU101 to set constraints on bacterial intake and oxygen consumption rates in the pFBA of GRU102. More specifically, we set the constraint on the bacterial intake rate of GRU102 by scaling the constraint of GRU101 according to the ratio of pharyngeal pumping rates between the two strains. Meanwhile, the oxygen consumption rate of GRU102 is constrained to a scalar multiple of the oxygen consumption rate of GRU101 as determined by pFBA, based on the ratio of experimentally observed oxygen uptake rates between the two strains.

## Flux Variability Analysis and inconsistent reactions

We performed flux variability analysis (FVA) (*Gudmundsson and Thiele, 2010*) on the GRU101 and GRU102 metabolic models to determine the range (interval) of flux values for each reaction, within which one can vary the flux of a reaction – keeping the others constant – without changing the optimal biological objective value. We implemented the FVA using in-built functions in the COBRA toolbox (*Schellenberger et al., 2011*). We compared the results of FVA of GRU101 and GRU102 to identify as inconsistent reactions. Here, a reaction is inconsistent between the two animals when its flux interval according to FVA is located fully to the left (negative) or right (positive) of the origin for at least one of the strains, and the flux intervals between the two animals have values of opposite signs.

## TCA enzyme activity assay

Mitochondrial extraction was carried out as described in *Goo et al. (2012)*. Mitochondria proteins were used for alpha-Ketoglutarate dehydrogenase (aKGDH), Succinyl-CoA Synthetase, (SCS), Malate dehydrogenase (MDH) activity assays while cytosolic protein was used for Lactate dehydrogenase (LDH) assay. All assays were performed using colometric assay kits (BioVision Inc, San Francisco, USA) according to manufacturer's instruction. 20 μg of mitochondrial protein was used for aKGDH assay (Cat. No K678). 10 μg of mitochondrial protein was used for SCS assay (Cat. No K597). 2 μg of mitochondrial protein was used for MDH assay (Cat. No K654). 10 ug of cytosolic protein was used for LDH (Cat. No K726) assay.

## Protein carbonyl assay

Protein carbonyl assay was performed as described in *Ng et al. (2014)*.

## Lipidomics analysis

Approximately, 1500 nematodes per sample were collected, washed with M9 buffer and transferred to 2 mL polypropylene tubes containing 250 μL lysis buffer (20 mM Tris-HCl pH 7.4, 100 mM NaCl,

0.5 mM EDTA, 5% glycerol) and left on ice for 15 min followed by homogenization using a Bead beater (Precellys, France) maintained at 4°C. Lipids extraction from the lysed samples was carried out by Folch's extraction. In order to minimize oxidation during and after extraction, 0.5% butylated hydroxytoluene (BHT) was added to the organic solvents. Samples were spiked with known amounts of internal standards (purchased from Avanti polar lipids, Alabaster, AL, USA) corresponding to each lipid class during the extraction to control lipid-class dependent differences in extraction and ionization. Internal standard used for triacylglyceride (TAG) was d5-TAG 48:0. To achieve separation into aqueous and organic phases, the samples were vortexed and centrifuged at 3000 rpm for 5 min. The lower phase was then transferred to a fresh centrifuge tube and centrifuged in a vacuum concentrator (SpeedVac, Thermo Savant, Milford, USA) until dry. The dried lipid extract was reconstituted in 50 μL methanol and kept at −80°C until mass spectrometry analysis.

## Lipidomics mass spectrometric analysis and data processing

A 6490 triple Quad Mass spectrometer (QqQ; Agilent, USA) coupled to a 1260-Ultra Performance Liquid chromatography (UPLC) was used for lipid quantification. ESI was used to ionize lipids. Each lipid molecular species was analyzed using a targeted multiple reaction monitoring (MRM) approach containing transitions for known precursor/product mass-to-charge ratio and retention times. The UPLC system was equipped with a Waters ACQUITY BEH C18 column (1.0 × 100 mm) to separate the molecular species using gradient elution. Solvent A was acetonitrile/$H_2O$ (60:40) with 10 mM ammonium formate while solvent B was isopropanol/acetonitrile (90:10) containing 10 mM ammonium formate. The flow rate was 0.4 mL/min and the column temperature 50°C. Solvent B was set at 70% at injection and increased linearly to 100% in 14 min, retained at this value for 3 min, decreased to 10% in one minute and then retained there until the end of the gradient by 14 min. The eluent was directed to the ESI source of the mass spectrometer operated in the positive ion mode. The MS conditions were as follows. For ESI: gas temperature, 250°C; gas flow, 14 L/minutes; sheath gas temperature, 250°C; sheath gas flow, 11 L/minutes; and capillary voltage, 3,500 V.

LC-MS data obtained on the 6490 QqQ Mass spectrometer were subjected to data processing using MassHunter software (Agilent). The identification of a lipid species was based on mass and retention times (RT). Signal intensities of the targeted species were compared with the intensities from the spiked internal standards and the retention times for the various classes were matched. Data processing, included peak smoothing and integration of areas under the curves for each measured transition. The processed data were exported to excel and normalized to protein content as well as the internal standard. Fold changes were measured by comparison of the different conditions to the control sample and finally the Student's t test was performed to determine whether differences between the samples were statistically significant. ($p < 0.05$ was considered statistically significant).

## Protein aggregation assay

Protein aggregation assay was carried out as described in *David et al. (2010)* with some modifications. Worms were collected and washed twice in M9 buffer to remove excess bacteria. On the day of homogenization, worm pellet was re-suspended in RIPA buffer (50 mM Tris, 150 mM NaCl, 5 mM EDTA, 0.5% SDS, 0.5% Sodium Deoxycholate, 1% Triton X-100, 1X Protease inhibitors and 1 mM PMSF) and homogenized using a Bead Beater (Benchmark Scientific, Edison, USA). To obtain the detergent-soluble protein, the lysate was removed after the first extraction following centrifugation at 14,000 rpm. Sequential extraction was done by adding fresh RIPA buffer to the pellet, followed by re-homogenization in bead beater, and then removing the lysate and repeating this process for another four times. This was to ensure that all the detergent-soluble protein had been extracted. Soluble fraction from these wash steps were then pooled and the amount of detergent-soluble protein present in the pooled sample was determined using Bradford protein assay (DC Protein Assay Kit, Bio-rad Laboratories, California, United States). Loading volumes for soluble and insoluble protein fractions were scaled to protein content of this pooled soluble fraction.

To obtain the detergent-insoluble protein, 8M Urea lysis buffer (8M Urea, 1 mM DTT, 50 mM Tris, 2% SDS, 1x Protease inhibitors and 1x Phosphatase inhibitors) was added to the pellet, vortexed for 10 s, centrifuged at 14,000 rpm for 10 min at room temperature and the lysate was collected. This process was repeated using fresh Urea buffer for five times in total. 10 ul of lysate from each of the

urea extraction was then pooled to form the pooled insoluble protein for loading on the SDS-PAGE gel. For the SDS-PAGE gel, the volume loaded for each insoluble protein sample was five times the volume equivalent of 2 µg of soluble protein determined in the pooled soluble protein lysate earlier. SDS-PAGE gel containing 0.5 µg of soluble protein lysate loaded for each sample was further performed to normalize for any differences in protein loading. SDS-PAGE gel was run using a BioRad gel electrophoresis system. Upon completion of gel electrophoresis, total protein amount was visualized using Silver Stain Kit (Cat. No 24612, Thermo Fisher Scientific, Massachusetts, USA). Densitometric analysis of gel intensity was performed using ImageJ.

### Lifespan assay

Lifespan assay was conducted under blinded condition as described in *Ng et al. (2014)* in the presence of 250 uM FUDR.

### Paraquat challenge

Paraquat (PQ) challenge was done in a 96-well plate liquid medium. Animals were raised in NGM plates without PQ until day 10 of age. Subsequently, day 10 animals were transferred to M9 buffer containing 100 mM PQ. The number of dead animals was then monitored every 30 min.

### Metabolic flux assay

Metabolic flux assay was conducted using Seahorse Bioanalyzer as described in *Fong et al. (2017)*.

### MitoSox staining

MitoSox staining was carried out as described in *Teo et al. (2018)*. Briefly, day three animals were grown on NGM plates containing 10 µM MitoSox (Molecular Probes, Singapore). Following 24-hr incubation, animals were transferred to fresh NGM for 1 hr. Thereafter, animals were imaged using a confocal microscope (Zeiss LSM800) at 20X magnification, at a wavelength of 540/25 nm.

### MitoTracker staining

MitoTracker staining was performed as described in *Gaffney et al. (2014)*. Briefly, 20 nematodes were incubated with 20 µl of 4.7 µM MitoTracker Red (Thermo Fisher Scientific, Waltham, United States). Following 1 hr incubation, nematode suspension was washed in M9 buffer for three times. Animals were then imaged using a confocal microscope (Zeiss LSM800) at 20X magnification, at a wavelength of 540/25 nm.

## Acknowledgements

Some strains were provided by the CGC, which is funded by NIH Office of Research Infrastructure Programs (P40 OD010440). We thank the technicians in the metabolomics facility for running the samples. We are grateful for the funding provided by Ministry of Education Singapore (Grant R-184-000-230-112, 2014-T2-2-120), Yale-NUS grant IG17-LR006, Swiss National Foundation (Grant #163390) and NUS Graduate School for Integrative Sciences and Engineering.

## Additional information

### Funding

| Funder | Grant reference number | Author |
|---|---|---|
| Ministry of Education - Singapore | NGS Scholarship | Emelyne Teo |
| Ministry of Education - Singapore | R-184-000-230-112 | Jan Gruber |
| Swiss National Science Foundation | 163390 | Sudharshan Ravi |
| Yale-NUS | IG17-LR006 | Emelyne Teo Nicholas Tolwinski |

| Ministry of Education - Singapore | 2014-T2-2-120 | Jan Gruber |

The funders had no role in study design, data collection and interpretation, or the decision to submit the work for publication.

## Author contributions

Emelyne Teo, Conceptualization, Data curation, Formal analysis, Investigation, Methodology, Writing—original draft, Writing—review and editing; Sudharshan Ravi, Diogo Barardo, Amaury Cazenave-Gassiot, Tsze Yin Tan, Jianhong Ching, Jean-Paul Kovalik, Markus R Wenk, Rudiyanto Gunawan, Data curation, Formal analysis, Methodology, Writing—review and editing; Hyung-Seok Kim, Methodology, Writing—review and editing; Sheng Fong, Philip K Moore, Barry Halliwell, Data curation, Formal analysis, Writing—review and editing; Nicholas Tolwinski, Data curation, Formal analysis, Funding acquisition, Writing—review and editing; Jan Gruber, Conceptualization, Data curation, Formal analysis, Supervision, Funding acquisition, Investigation, Methodology, Writing—original draft, Project administration, Writing—review and editing

## Author ORCIDs

Emelyne Teo (iD) http://orcid.org/0000-0001-5050-4109
Diogo Barardo (iD) http://orcid.org/0000-0001-6177-0728
Jean-Paul Kovalik (iD) http://orcid.org/0000-0003-3654-8193
Jan Gruber (iD) https://orcid.org/0000-0003-3329-3789

## Decision letter and Author response

Decision letter https://doi.org/10.7554/eLife.50069.032
Author response https://doi.org/10.7554/eLife.50069.033

# Additional files

## Supplementary files

• Supplementary file 1. Nutritional and metabolic parameters used in metabolic flux balance analysis.
DOI: https://doi.org/10.7554/eLife.50069.026

• Supplementary file 2. List of reactions that show directional inconsistency in flux variability analysis (FVA).
DOI: https://doi.org/10.7554/eLife.50069.027

• Supplementary file 3. Summary of all lifespan trials with Metformin-treatment.
DOI: https://doi.org/10.7554/eLife.50069.028

• Transparent reporting form
DOI: https://doi.org/10.7554/eLife.50069.029

## Data availability

All data generated or analysed during this study are included in the manuscript and supporting files.

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
