## [Decision Letter]

Thank you for submitting your article "Metabolic stress is a primary pathogenic event in transgenic *Caenorhabditis elegans* expressing neuronal human amyloid-beta " for consideration by *eLife*. Your article has been reviewed by three peer reviewers, including Matt Kaeberlein as the Reviewing Editor and Reviewer #1, and the evaluation has been overseen by Jessica Tyler as the Senior Editor. The following individuals involved in review of your submission have agreed to reveal their identity: George L Sutphin (Reviewer #2); Michael Petrascheck (Reviewer #3).

The reviewers have discussed the reviews with one another and the Reviewing Editor has drafted this decision to help you prepare a revised submission.

Summary:

Teo et al. present a study examining the role of impaired mitochondrial metabolism in amyloid-beta neuropathology in transgenic *C. elegans*. Specifically, they examine the link between early changes in the TCA cycle and late changes in proteostasis and related pathology in a previously published strain with relatively low-level transgenic pan-neuronal expression of human amyloid-beta that results in late-onset amyloid pathology.

To begin the analysis, they used metabolomics to examine intermediate metabolites related to the TCA cycle and several pathways that provide precursor molecules. These results suggested that some aspects of TCA metabolism were perturbed in amyloid-beta animals. The authors then performed flux analysis using a combination of transcriptomic and metabolomic data to determine which TCA-associated pathways were specifically altered in these animals, identifying the conversion of pyruvate to alpha-ketoglutarate (aKG) as a major focal point. They next identified an age-dependent decrease in aKG dehydrogenase (aKGDH), and demonstrate that RNAi knockdown of aKGDH in non-amyloid-beta transgenic controls mimics several of the metabolic defects (e.g. oxygen consumption profiles) in amyloid-beta animals, suggesting that this may be a mediating factor in pathology driven by metabolic differences. They propose oxidative damage as a potential driver of mitochondrial dysfunction, and report elevated protein carbonylation specifically in the mitochondria. Finally, they demonstrate that metformin rescues both survival deficits and metabolic dysfunction in amyloid-beta animals.

This work adds valuable insight into early metabolic events in amyloid-beta neurotoxicity in a malleable model system. We believe the paper will be of broad interest to the scientific community, particularly those working in, or in areas adjacent to, aging and Alzheimer's disease. Metabolic dysfunction as an early even in AD is of particular interest, as is the identification of aKGDH as a potential focal point and potential intervention target. We have identified a few significant concerns that the authors should address in a revised submission.

Essential revisions:

One concern is whether the reported effects are primarily confined to the neurons or if they are cell non-autonomous effects caused by neuronal dysfunction. The data do not address this question, which to some extent is central to the model that metabolic defects happen before protein aggregation. This also impinges on the ROS model, since presumably ROS would be largely confined to neurons. While perhaps not absolutely essential if addressed thoroughly in the text, if the authors could provide some data clarifying the cell-autonomous versus cell non-autonomous nature of their observations that would greatly strengthen the manuscript. Related to this, unc-119 has been suggested to have some low expression in the gut.

There are a couple of concerns related to the effects of metformin.

1) The effects seem variable and perhaps interpreted too strongly. For example, the text states that "We found that Metformin treatment indeed normalized total levels of insoluble protein in GRU102 to that of GRU101 animals (Figure 6A, B)." Looking at panel A, two of the four Metformin treated samples look indistinguishable from age-matched GRU102, and the other two look indistinguishable from GRU101. How was protein loading measured/normalized/verified (there is a hint in the Materials and methods that protein concentration of the soluble fraction may have been measured, but no detail)? In panel B, the comparisons indicate that the Metformin group is not statistically different from either. These patterns in the data complicate the interpretation. The text needs to reflect the data here, and the strength of the claims should be placed in context of the variability of the Metformin data. This claim is also made in the legend of Figure 7 and in the Discussion.

2) The mechanism of action for metformin is unclear. Metformin is an inhibitor of Complex I of the ETC and has also been shown to impact lifespan in *C. elegans* through inhibition of folate production by the microbiome. Either (or neither) could be contributing to the observed effects in GRU102. Effects of bacterial folate limitation on lifespan are supported by published data from several labs that sulfa antibiotics can increase lifespan in worms. It would enhance the impact of this manuscript if the authors explored whether the effects of metformin are microbiome mediated. Two easy ways to do this would be to (1) determine whether sulfa antibiotics phenocopy metformin in the AD worms and (2) determine whether metformin still has effects when killed bacterial food is used for the assays.

3) Please provide data and statistics for all independent replicates indicated in the text and figure legends as supplementary material.

---

## [Author Response]

Essential revisions:One concern is whether the reported effects are primarily confined to the neurons or if they are cell non-autonomous effects caused by neuronal dysfunction. The data do not address this question, which to some extent is central to the model that metabolic defects happen before protein aggregation. This also impinges on the ROS model, since presumably ROS would be largely confined to neurons. While perhaps not absolutely essential if addressed thoroughly in the text, if the authors could provide some data clarifying the cell-autonomous versus cell non-autonomous nature of their observations that would greatly strengthen the manuscript. Related to this, unc-119 has been suggested to have some low expression in the gut.

We agree with the reviewers that it would be important to clarify cell-autonomous vs. non-autonomous mechanisms underlying the metabolic defects observed in GRU102. We have attempted to answer the question by performing whole body mitoSOX and mitoTracker staining of the animals, to respectively examine ROS localization and mitochondrial morphology in neuronal versus non-neuronal compartments of the animals. However, we found that the mitoSOX dye predominantly localizes at the pharyngeal bulb as also previously reported^1^, and the resolution we were able to get was not high enough to clearly differentiate between neuronal and other cellular structure (Figure 1A) and we did not see a clear difference in gut (as might be expected by unc-119 driven leaky expression). Similarly, we also could not distinguish between neuronal vs. other cellular compartment from the mitotracker stain (Figure 1B).

We have added these data to the manuscript, briefly mentioning it as part of the overall discussion. However, we appreciate that these data are not sufficient to fully address this point. In line with the reviewers’ suggestion, we have therefore added a more complete discussion of the question and clearly stated this limitation in our current understanding as follows:

Discussion:

“It is intriguing that GRU102 animals displayed such broad metabolic changes, given that the Aβ transgene is only expressed in neurons. […] While our data currently do not address this question, the Aβ-induced oxidative stress hypothesis of AD (see Introduction paragraph two) supports oxidative stress as a direct and cell-autonomous mechanism of Aβ-induced toxicity^2^. The confined effects of oxidative stress in Aβ-expressing neurons is also supported by clinical findings that ROS-mediated elevation in PCC are only observed in brain regions of AD subjects rich in Aβ-plaques, but not in other brain regions devoid of Aβ-plaques^3^.

On the other hand, existing literature has suggested the activation of stress response mechanism and proteotoxicity as non-cell-autonomous consequences in protein-misfolding diseases including AD^4^, as activation of stress response, such as the heat shock response or unfolded protein response, functions at a whole animal level, beyond the immediate affected cell^4^. Studies in *C. elegans* have also shown that induction of mitochondrial stress in neurons can lead to non-cell-autonomous mitochondrial stress response in non-neuronal compartment such as intestine^5^. The mechanism of the impairment in mitochondrial health observed in GRU102 may therefore not be confined to neuronal cells only. The pan-neuronal unc-119 promoter that we use to drive Aβ expression in GRU102 has been suggested to have low expression in intestine, and such expression might lead to detriments in that compartment not confined in the neurons.”

There are a couple of concerns related to the effects of metformin.1) The effects seem variable and perhaps interpreted too strongly. For example, the text states that "We found that Metformin treatment indeed normalized total levels of insoluble protein in GRU102 to that of GRU101 animals (Figure 6A, B)." Looking at panel A, two of the four Metformin treated samples look indistinguishable from age-matched GRU102, and the other two look indistinguishable from GRU101. How was protein loading measured/normalized/verified (there is a hint in the Materials and methods that protein concentration of the soluble fraction may have been measured, but no detail)? In panel B, the comparisons indicate that the Metformin group is not statistically different from either. These patterns in the data complicate the interpretation. The text needs to reflect the data here, and the strength of the claims should be placed in context of the variability of the Metformin data. This claim is also made in the legend of Figure 7 and in the Discussion.

We agree with the reviewers that the effect of Metformin on protein aggregation is variable and we apologize for missing out some of the details on protein loading and normalization. In response to the reviewers’ comments, we have re-analysed this dataset, generated additional normalization data and performed additional validation experiments.

Briefly, the amount of insoluble protein loaded was based on the amount of soluble protein in the soluble fraction, where the volume for each insoluble protein sample loaded was five times the volume of soluble lysate, equivalent to 2ug soluble protein as determined by Biorad Bradford protein assay (DC Protein Assay Kit). In other words, the amount loaded was normalized to the protein content of the soluble fraction as evaluated by Biorad Bradford protein assay (DC Protein Assay Kit). We have further clarified this in the Materials and methods section as follows:

“To obtain the detergent-soluble protein, the lysate was removed after the first extraction following centrifugation at 14,000rpm. […] Upon completion of gel electrophoresis, total protein amount was visualized using Silver Stain Kit (Cat. No 24612, Thermo Fisher Scientific, Massachusetts, USA). Densitometric analysis of gel intensity was performed using ImageJ.”

Following the reviewers’ comments, we also re-examined the volume normalization process to explore reasons for the observed variability in insoluble protein seen in the metformin-treated cohorts. We considered the possibility that the variability was due to inaccuracies in quantification of the soluble protein fraction, resulting in increased variability due to loading volumes based on this quantification. We typically keep these soluble fractions but do not always run them on SDS-PAGE. We have now followed up this question by running these soluble protein fractions on SDS-PAGE gels. The amount of insoluble protein loaded for comparison is based on the assumption that the amount soluble protein in each lane should be precisely 0.2μg. We only had soluble protein fractions for a second replicate of the Metformin trials, as we did not retain the soluble protein samples for the first batch of samples originally shown in the manuscript. We therefore ran SDS-PAGE of the soluble protein fractions from the second experiments and replaced the graphs and gels in the manuscript with this second replicate. For statistical analysis, we now re-normalize data of insoluble protein using the densitometric intensity of the SDS-PAGE of the corresponding soluble fractions (Figure 6). We have moved the original unnormalized gel to the supplementary data (Figure 6—figure supplement 1).

We found that the actual amount of soluble protein indeed varied slightly between samples (Figure 6B). While the somewhat reduced variability in normalized values confirms that some of the observed variability was likely due to loading volume, the normalization did not change any of the overall statistics and conclusions. After correcting for the loading differences by normalizing the densitometric intensity of insoluble gel against that of the soluble gel, the result remained qualitatively the same as the previously unnormalized data. Even though Metformin treatment showed a trend of reduced protein aggregates level in D12 GRU102, the reduction was not statistically significant (Figure 6C, One-way ANOVA multiple comparisons test P=0.08). However, the levels of protein aggregates in D12 Metformin-treated GRU102 also appear similar to the D12 GRU101 controls (Figure 6C, One-way ANOVA multiple comparisons test P=ns), and the levels of protein aggregates in the Metformin-treated animals were intermediate between the untreated GRU101 and GRU102. We therefore felt that there were some benefits in reducing protein aggregates conferred by Metformin, even though it does not fully rescue it.

We therefore agree with the reviewers that care should be taken with regards to claims made regarding this effect. We apologize for not more strongly emphasize this in the original version of the manuscript. We agree that we cannot conclude unambiguously that the effects of Metformin on protein aggregation due to the variability of the gel and hence suggest the following modification of our claim:

Results:

“To determine whether general protein aggregation occurs in the Aβ-expressing animals, we examined the levels of insoluble proteins *via* SDS-PAGE^6^. […] Metformin treatment therefore appears to benefit GRU102 in terms of proteostasis but it did not fully rescue the increase in protein aggregates.”

Figure legend:

**“**Figure 7. Graphical summary of the model of Aβ pathology in GRU102. […] Metformin-treated GRU102 also appears to reduce the increase in protein aggregates in GRU102, even though it could not fully rescue this phenotype.”

Discussion:

“We further found that Metformin-treated GRU102 showed a trend towards lower levels of protein aggregates. […] While more work needs to be carried out to confirm the causal relationship between metabolic dysfunction and protein aggregation, our observation is consistent with findings that metabolic health and proteostasis are closely related to each other^7-10^.”

2) The mechanism of action for metformin is unclear. Metformin is an inhibitor of Complex I of the ETC and has also been shown to impact lifespan in *C. elegans* through inhibition of folate production by the microbiome. Either (or neither) could be contributing to the observed effects in GRU102. Effects of bacterial folate limitation on lifespan are supported by published data from several labs that sulfa antibiotics can increase lifespan in worms. It would enhance the impact of this manuscript if the authors explored whether the effects of metformin are microbiome mediated. Two easy ways to do this would be to (1) determine whether sulfa antibiotics phenocopy metformin in the AD worms and (2) determine whether metformin still has effects when killed bacterial food is used for the assays.

We thank the reviewers for suggesting these experiments and agree that conducting this work would increase the value of the study. In response to the reviewers comments, we have performed the first suggested experiment – to determine whether Sulfamethoxazole (SMX), a sulfonamide antibiotics that has been previously shown to reduce folate levels and increase lifespan in worms (de la Monte et al., 2008), results in a rescue in metabolic flux impairment and/or lifespan in GRU102. These data suggest that inhibition of microbiome folate production through SMX does not have any beneficial effects in GRU102, hence lessening the probability that the benefits of Metformin were predominantly microbiome-mediated. We have added these results as follow:

Results:

“Metformin has been previously shown to impact lifespan in *C. elegans* through inhibition of folate production by the microbiome. […] These observations suggest that the beneficial effects of Metformin in GRU102 were unlikely to be mediated predominantly by inhibition of microbiome folate production.”

3) Please provide data and statistics for all independent replicates indicated in the text and figure legends as supplementary material.

We apologize for missing out those data and have now provided them in full as Figure 1—figure supplement 1, Figure 5—figure supplement 1, and Supplementary file 3.

For all metabolomics data, we have also taken the opportunity to do a detailed comparison for results for each individual metabolite from the two repeats. As indicated, overall the data and statistics for these two independent replicates are consistent. However, there are four metabolites that show minor differences, and we have now also explicitly listed and discussed these. These differences concern the acylcarnitine and amino acids metabolites levels, where some of the originally reported changes did not reach statistical significance in the second replicate. However, the general pattern that we observe clearly holds in both datasets, and none of these differences affect the major claim that we made in the manuscript. We have nevertheless modified the text accordingly to reflect the differences in the independent replicate and report only the consensus input following Author response table 1:

**Author response table 1. resptable1:** 

	Original data in manuscript	Data in independent replicate	Modifications made in the manuscript
Figure 1A	Acylcarnitines derived from amino acids, C3 and C5, were significantly reduced in D12 GRU102	Only C3 was significantly reduced in D12 GRU102; C5 showed a trend towards reduction but the difference did not reach statistical significance in D12 GRU102	We removed the sentence stating that C5-Acylcarnitine was significantly reduced and focused only on C3-Acylcarntine: “AA-derived AC species (C3-carnitine) was also significantly reduced in old GRU102 (Figure 1A), suggesting that AA metabolism was affected in old GRU102.”
Figure 1D	Percentage of amino acids forming pyruvate were significantly lower in both D4 and D12 GRU102 Percentage of amino acids forming aKG were significantly higher in both D4 and D12 GRU102	Reduction in percentage of amino acids forming pyruvate did not reach statistical significance in both D4 and D12 GRU102 Increase in percentage of amino acids forming aKG was only statistically significant in D12 GRU102, but not in D4 GRU102	We removed the phrase that the percentage of amino acids forming pyruvate was significantly lower and focused only on the increase in percentage of amino acids forming aKG: “This analysis of the AA profile demonstrated that, in comparison to age-matched controls, the percentage of glucogenic AA forming alpha-Ketoglutarate (aKG) was significantly increased in old GRU102 animals (Figure 1D).”
Figure 5D	Metformin significantly increased levels of Ala, Asp, Gly, Leu, Met, Phe, Pro, Ser, Trp and Val in D12 GRU102.	Metformin significantly increased levels of Ala, Asp, Gly, Phe, Pro, Ser while significantly decreased levels of Arg and Tyr in D12 GRU102.	The general trend that Metformin increased levels of several individual amino acids still holds, even though the two replicates showed a slight variation in the identity of amino acids that were increased. We have modified our wording as follows: “We found that Metformin treatment increased the abundance of many AA in GRU102, with Alanine, Glycine, Proline and Serine consistently increased in both replicates (Figure 5D, Figure 5—figure supplement 1).”
Figure 5E	Metformin significantly increased levels of C3-acylcarnitine and also C5-acylcarnitine, even though the latter did not reach statistical significance.	Levels of C3-acylcarnitine appear similar between Metformin-treated and untreated GRU102. Metformin also appears to reduce the levels of C5-acylcarnitine, even though the reduction did not reach statistical significance.	Given the different trends observed in the two replicates, we feel that the increase in C3-acylcarnitine originally observed was within the noise of the measurement, and hence has removed its discussion in the manuscript.

References:

1) Dingley, S. et al. Mitochondrial respiratory chain dysfunction variably increases oxidant stress in *Caenorhabditis elegans*. Mitochondrion 10, 125-136, doi:10.1016/j.mito.2009.11.003 (2010).

2) Butterfield, D. A. and Halliwell, B. Oxidative stress, dysfunctional glucose metabolism and Alzheimer disease. Nat Rev Neurosci 20, 148-160, doi:10.1038/s41583-019-0132-6 (2019).

3) Hensley, K. et al. Brain regional correspondence between Alzheimer's disease histopathology and biomarkers of protein oxidation. J Neurochem 65, 2146-2156, doi:10.1046/j.1471-4159.1995.65052146.x (1995).

4) Nussbaum-Krammer, C. I. and Morimoto, R. I. *Caenorhabditis elegans* as a model system for studying non-cell-autonomous mechanisms in protein-misfolding diseases. Dis Model Mech 7, 31-39, doi:10.1242/dmm.013011 (2014).

5) Durieux, J., Wolff, S. and Dillin, A. The cell-non-autonomous nature of electron transport chain-mediated longevity. Cell 144, 79-91, doi:10.1016/j.cell.2010.12.016 (2011).

6) David, D. C. et al. Widespread protein aggregation as an inherent part of aging in *C. elegans*. PLoS Biol 8, e1000450, doi:10.1371/journal.pbio.1000450 (2010).

7) Ruan, L. et al. Cytosolic proteostasis through importing of misfolded proteins into mitochondria. Nature 543, 443-446, doi:10.1038/nature21695 (2017).

8) Patel, A. et al. ATP as a biological hydrotrope. Science 356, 753-756, doi:10.1126/science.aaf6846 (2017).

9) Fang, E. F. et al. Mitophagy inhibits amyloid-beta and tau pathology and reverses cognitive deficits in models of Alzheimer's disease. Nat Neurosci 22, 401-412, doi:10.1038/s41593-018-0332-9 (2019).

10) Guerrero-Gomez, D. et al. Loss of glutathione redox homeostasis impairs proteostasis by inhibiting autophagy-dependent protein degradation. Cell Death Differ, doi:10.1038/s41418-018-0270-9 (2019).

11) Virk, B. et al. Excessive folate synthesis limits lifespan in the *C. elegans*: *E. coli* aging model. BMC Biol 10, 67, doi:10.1186/1741-7007-10-67 (2012).